# Intrinsic Gradient Suppression for Label-Noise Prompt Tuning in Vision–Language Models

Jiayu Li [* 1]   Jiaxin Qi [* 3]   Sheng Zhou [1]   Jianqiang Huang [2 3 4]   Xiansheng Hua [2]

## Abstract

Contrastive vision-language models like CLIP exhibit remarkable zero-shot generalization. However, prompt tuning remains highly sensitive to label noise, as mislabeled samples generate disproportionately large gradients that can overwhelm pre-trained priors. We argue that because CLIP already provides a near-optimal initialization, adaptation should be inherently conservative, particularly against the extreme gradient updates common in noisy settings. To this end, we propose Double-Softmax Prompt Tuning (DSPT), a hyperparameter-free method for intrinsic gradient suppression. By applying a sequential probabilistic normalization, DSPT induces a self-adaptive saturation zone that suppresses gradients from high-error noisy samples while maintaining informative updates. We also provide both theoretical analysis and empirical evidence about how this mechanism achieves adaptive suppression. This design transforms "gradient vanishing", traditionally a training bottleneck, into a principled noise-filtering shield for label-noise prompt tuning. Extensive experiments confirm that this simple, drop-in design achieves state-of-the-art robustness across various noisy benchmarks, outperforming methods with complex architectures and handcrafted hyperparameters.

## 1. Introduction

Contrastively trained large-scale vision–language models, such as CLIP (Radford et al., 2021), align images and captions in a shared embedding space by pulling matched im-

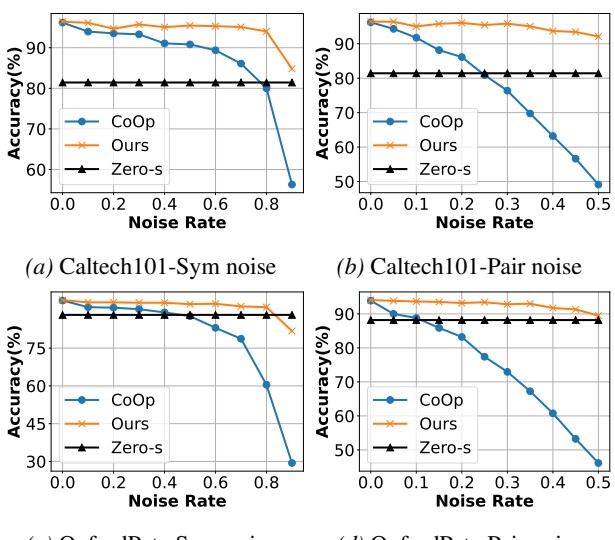

*(a) Caltech101-Sym noise*   *(b) Caltech101-Pair noise*

*(c) OxfordPets-Sym noise*   *(d) OxfordPets-Pair noise*

*Figure 1.* Accuracy curve of prompt-tuning(CoOp), our method, and zero-shot predictions with increasing noise rate under different settings. CoOp suffers from a significant performance drop and falls below zero-shot predictions, while our double-softmax cross-entropy loss yields consistent noise robustness.

age–caption pairs together and pushing negatives apart. This alignment enables zero-shot image classification, where class names are inserted into prompts (e.g., "a photo of a [class]") and the model can predict the class whose prompted text embedding is most similar to the image embedding without training on the target images. To improve classification performance while mitigating overfitting, CLIP-specific prompt tuning (Zhou et al., 2022b) keeps the backbone frozen and learns a set of continuous tokens, $[v_i], i = 1, 2, \ldots, l$, that are concatenated to the class name (e.g., "$[v_1] [v_2] \ldots [v_l]$ [class]"), producing task-specific textual prototypes better aligned with the target images. For example, in fine-grained car classification, the learned prompts can implicitly emphasize car-specific attributes (e.g., grille, headlight shape), thereby improving image–text alignment and classification accuracy.

However, the presence of label noise severely disrupts this adaptation. In standard prompt tuning, the cross-entropy objective generates disproportionately large gradients for mislabeled samples, allowing them to dominate the adap-

*Equal contribution [1]Zhejiang University, Hangzhou, China [2]Tongji University, Shanghai, China [3]Computer Network Information Center, Beijing, China [4]University of Chinese Academy of Sciences, Beijing, China. Correspondence to: Jianqiang Huang <jianqiang.jqh@gmail.com>.

*Proceedings of the 43rd International Conference on Machine Learning*, Seoul, South Korea. PMLR 306, 2026. Copyright 2026 by the author(s).

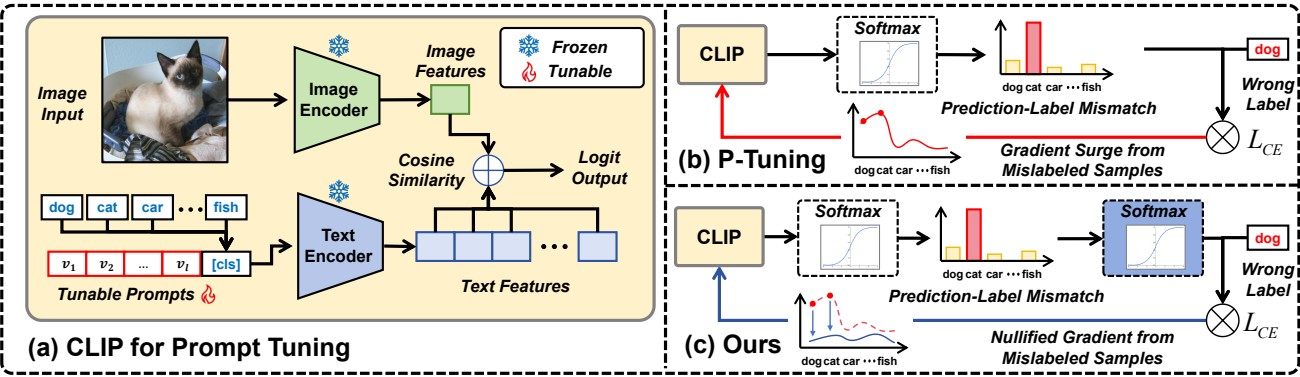

*Figure 2.* Overview of the proposed method. (a) CLIP backbone model for prompt-tuning. (b) We observe that prompt-tuning generates gradient surges in samples with confident predictions inconsistent with noisy labels with respect to output logit terms, which are likely to be mislabeled and will therefore harm the training process. (c) Our double-softmax cross-entropy loss nullifies the influence of mismatch samples by suppresing gradient, which mitigates overall overfitting.

tation updates. For example, when an image of a "car" is mislabeled as a "dog", CLIP's strong prior correctly identifies the mismatch, resulting in a near-zero prediction probability for the noisy label. Paradoxically, since the gradient magnitude of the cross-entropy loss is proportional to the residual between the ground truth label and the prediction, the more "incorrect" a label appears to the pre-trained prior, the larger the gradient surge it produces. These surges aggressively overrule the pre-trained priors, forcing the model to rapidly fit errors and leading to progressive performance degradation. As shown in Figure 1, the model eventually falls below the zero-shot baseline, rendering vanilla prompt tuning detrimental. This creates a fundamental conflict between CLIP's reliable prior and label noise, motivating a more principled denoising design for prompt tuning.

Given that CLIP already provides a near-optimal starting point, we argue that the adaptation process should be inherently conservative, prioritizing the preservation of original knowledge over aggressive reconfiguration. In this paper, we propose Double-Softmax Prompt Tuning (DSPT), a hyperparameter-free framework designed for intrinsic gradient suppression for pre-trained models' label-noise prompt tuning. Our core insight is that for pre-trained models like CLIP, "gradient vanishing", traditionally a critical defect in training from scratch, can be re-purposed as a principled noise-filtering shield. By applying two sequential probabilistic normalizations, DSPT induces a self-adaptive saturation zone that targets the extreme gradients produced by mislabeled samples. Specifically, for highly unreliable noisy samples, the double-softmax pushes logits into the saturation region, effectively "zeroing out" their high-magnitude gradients. Conversely, for informative samples, those where the model is uncertain but the signals are potentially reliable, the gradients remain within the active learning zone. While these informative gradients are also slightly tempered, they are permitted to pass through, enabling the subtle re-

finements necessary for robust CLIP-based prompt tuning. More details are in the theoretical analysis and Figure 3.

Different from current denoising methods, such as Logit-Norm (Wei et al., 2022) or LogitCLIP (Wei et al., 2023), which often rely on complex, handcrafted hyperparameters for logit clamping, DSPT offers a more elegant and robust solution. Existing methods typically employ static geometric constraints or global logit scaling that fail to distinguish between samples adaptively. Furthermore, they often require extensive tuning of hyperparameters that are highly sensitive to specific noise rates and datasets. In contrast, DSPT is an extremely simple, drop-in design that requires no hyperparameter tuning. Through in-depth experimental and theoretical analysis, we demonstrate that our intrinsic gradient suppression is far more effective for CLIP-based adaptation than complex, hand-designed corrections. Extensive experiments across various benchmarks confirm that DSPT consistently achieves state-of-the-art robustness, providing a simple yet powerful direction for label-noise prompt tuning in vision–language models

Our main contributions can be summarized as follows:

- We provide a systematic analysis of CLIP's gradient dynamics under label noise. We reveal a gradient surge paradox, where CLIP possesses a strong pre-trained prior and will produce disproportionately large gradients for "incorrect" noisy samples, which aggressively overrule the original knowledge.

- We propose Double-Softmax Prompt Tuning (DSPT), a hyperparameter-free framework that re-purposes "gradient vanishing" as a principled noise-filtering shield. By inducing a self-adaptive saturation zone, DSPT acts as an intrinsic gradient gatekeeper that zeros out disruptive surges from noise while selectively permitting informative updates from reliable samples.

- We provide theoretical evidence and conduct extensive experiments across diverse benchmarks. Our results show that this simple, drop-in design consistently achieves state-of-the-art robustness without requiring complex architecture or dataset-specific hyperparameter tuning, offering a new direction for robust vision-language model adaptation.

## 2. Related Works

### 2.1. Prompt Tuning for Vision-Language Models

Pre-trained vision-language models such as CLIP (Radford et al., 2021) have achieved promising results by aligning visual and linguistic features with initially hand-crafted prompts. However, identifying suitable prompts for specific downstream tasks remains a labor-intensive and non-trivial challenge. To address this issue, CoOp (Zhou et al., 2022b) has been proposed as an automatic prompt generator. CoOp model initializes continuous prompt vectors randomly and treats them as learnable parameters, which are then optimized using a few-shot training dataset. To improve CoOp's generalization ability on unseen categories, CoCoOp (Zhou et al., 2022a) introduces a small neural network that extracts visual information from image features and integrates it into the prompt, creating instance-dependent context for each image. BLIP (Li et al., 2022) introduces a unified encoder-decoder architecture that can easily transfer across different tasks. It also proposes a data augmentation approach that enhances the diversity of training data, improving model robustness. MaPLe (Khattak et al., 2023) designs a coupling function that generates visual prompts for the vision encoder from textual prompts. UPL (Huang et al., 2022) introduces unsupervised prompt tuning for VLMs by assigning pseudo-labels to unlabeled data, which are generated through the model's own zero-shot predictions. This method is further enhanced by Robust UPL (Wu et al., 2023), which incorporates random sample training and applies Generalized Cross Entropy (Zhang & Sabuncu, 2018) loss.

### 2.2. Learning with Noisy labels

Deep neural networks (DNNs) have been extensively studied and widely applied due to their powerful learning capabilities. However, this strength also makes them susceptible to training data with incorrect labels, leading to overfitting and performance degradation. Learning with noisy labels, or noisy label learning (NLL), focuses on mitigating the negative effects of label noise while preserving useful information during training. Common NLL approaches include selecting samples with correct labels (Li et al., 2020; Shu et al., 2019; Patel & Sastry, 2023; Wang et al., 2024; Zheng et al., 2021), correcting loss using intermediate information (Zheng et al., 2020; Yao et al., 2020; Xia et al., 2019), designing noise-robust model architectures (Reed et al., 2015;

Lukasik et al., 2020; Xia et al., 2021; Chen et al., 2021), and revising loss functions to be more resistant to noise (Ghosh et al., 2017b; Zhang & Sabuncu, 2018; Ma et al., 2020; Zhou et al., 2021). Notably, LogitClip (Wei et al., 2023) and LogitNorm (Wei et al., 2022) introduce robust loss functions that constrain the magnitude of the logit vector, aiding in noisy label learning and out-of-distribution detection.

To study the effect of label noise in VLM fine-tuning, Wu et al. (Wu et al., 2023) conduct experiments testing various fine-tuning methods with the presence of noisy labels, drawing the conclusion that prompt tuning is more noise-resistant compared to other fine-tuning strategies. Inspired by Wu et al., JoAPR (Guo & Gu, 2024) proposes a label-noise prompt-tuning approach by modeling the loss of training data with Gaussian Mixture Model(GMM) to form a self-adaptive threshold that distinguishes clean and noisy samples. Label correction is subsequently applied to mislabeled data through data augmentation and mixup (Zhang et al., 2018). Fang et. al (Fang et al., 2025) also utilize GMM-based sample selection for prompt tuning, while adopting class-specific prompts for label rectification. However, these two methods both introduce complicated mechanisms and involve time-consuming algorithms (retraining and BLIP model, respectively). NLPrompt (Pan et al., 2025) employs optimal transport to formulate pseudo label for sample selection, and introduces MAE loss for low confidence data.

## 3. Methodology

### 3.1. Preliminaries

**CLIP-Based Prompt Tuning**. Rather than relying on hand-crafted textual templates, prompt tuning learns a sequence of continuous prompt embeddings. For example, CoOp (Zhou et al., 2022b) introduces a shared learnable prompt for all classes, parameterized as $v_\theta = [v_1, v_2, \ldots, v_l]$ of length $l$, which is concatenated to each class embedding $e_c$. Given the image classification dataset $\mathcal{D} = \{(x_i, y_i)\}_{i=1}^n$, where $y_i \in \{1, \ldots, C\}$ is the ground-truth index, and a pre-trained CLIP model $f = \{f_{\text{img}}, f_{\text{text}}\}$, the objective for prompt tuning can be written as the standard cross-entropy loss:

$$z_{i,c} = f_{\text{img}}(x_i) \cdot f_{\text{text}}([v_\theta; e_c]),$$
$$\mathcal{L}_{\text{pt}} = -\frac{1}{n} \sum_{i=1}^n \log\big(\text{softmax}(z_i)_{y_i}\big) \tag{1}$$

where $[v_\theta; e_c]$ is the concatenated embedding for class $c$, $\cdot$ is the inner product, $z_i$ is the logits over $C$ classes for sample $i$, and $\text{softmax}(z_i)_{y_i} = \frac{\exp(z_{i,y_i})}{\sum_c \exp(z_{i,c})}$ denotes the predicted probability for the ground-truth class $y_i$.

**Noisy Label Classification**. Consider the same classification task under noisy labels. Let $\tilde{\mathcal{D}} = \{(x_i, \tilde{y}_i)\}_{i=1}^n$ denote

the corrupted training set, where $\tilde{y}_i \in \{1, \ldots, C\}$ is a potentially incorrect label obtained from the clean label $y_i$. We introduce the corruption process characterized by a transition matrix $T \in \mathbb{R}^{C \times C}$ with entries:

$$T_{jk} = \Pr(\tilde{y}_i = j \mid y_i = k), \tag{2}$$

so that the distribution of the noisy label depends only on the true class and is independent of $x_i$.

Instance-independent label noise is typically instantiated in two forms: (1) *Symmetric noise*, where each true label can be flipped to any other class with equal probability: $\forall k, T_{kk} = 1 - \eta, T_{jk} = \frac{\eta}{C-1}, j \neq k$, where $\eta \in [0, 1]$ is the noise rate. (2) *Asymmetric noise*, where the corruption is biased toward semantically similar classes. In practice, we adopt the common pair-flip model to simulate asymmetric noise, where each class can only be flipped to one particular other class: $\forall k, T_{kk} = 1 - \eta, \exists j \neq k, T_{jk} = \eta$, and all remaining off-diagonal entries are zero.

Formally, the goal in CLIP-based noisy label classification is to fine-tune the pre-trained CLIP on the noisy dataset $\tilde{\mathcal{D}}$ to outperform its zero-shot counterpart. As discussed above, prompt tuning provides some robustness to label noise, but its performance still degrades severely under high noise rates, motivating a more principled denoising method.

### 3.2. Our Method

Motivated by the insight that CLIP's strong priors cause noisy labels to generate disproportionately large gradients, we propose the Double-Softmax Prompt Tuning (DSPT) method for CLIP-based noisy prompt tuning:

$$\mathcal{L}_{\text{ours}} = -\frac{1}{n} \sum_{i=1}^{n} \log\big(\text{softmax}(\text{softmax}(z_i))_{\tilde{y}_i}\big), \tag{3}$$

where $z_i$ denotes prompt-tuned logits and $\tilde{y}_i$ denotes the noisy label. Compared to the standard objective in Eq. (1), our formulation introduces a sequential softmax normalization prior to the final prediction. Despite this minimalist design, the method induces an intrinsic gradient suppression mechanism, which adaptively targets and suppresses the extreme gradient surges typical of mislabeled samples, while allowing informative samples within the active learning zone to propagate.

In the following sections, we provide theoretical and empirical analyses demonstrating that this simple formulation effectively induces a gradient saturation zone, acting as an intrinsic filter against label noise.

### 3.3. Theoretical Analysis

Cross-entropy loss is widely applied for classification, but it is known to be fragile under label noise. Ghosh et al. (Ghosh

et al., 2017a) show that, as an asymmetric loss, cross-entropy is less noise-robust than symmetric losses such as MAE. Moreover, for a model $f_\theta$, the gradient of the per-sample cross-entropy loss with respect to the logit $z$ is:

$$\frac{\partial \mathcal{L}_{\text{pt}}(f_\theta(x), \tilde{y})}{\partial z_i} = p_i - \delta_{\tilde{y}i} \tag{4}$$

where $p = softmax(z)$ and $\delta_{ij}$ is the kronecker delta. For a model that gives inhomogeneous prediction on a noisy sample, in other words, $argmax_i(z_i) \neq \tilde{y}$, cross-entropy loss accumulates large gradients on $z_i$ and $z_{\tilde{y}}$, causing major disturbance for model training. This effect is amplified for high-confidence pre-trained vision-language models in prompt tuning, where large logits produce extremely sharp distributions, so mismatched samples induce larger gradients than correctly labeled samples. Since pre-trained vision-language models can give relatively accurate predictions, these samples are likely to be mislabeled and will mislead the model in the training process.

In the following, we show that the proposed double-softmax prompt-tuning mechanism can restrict the overall gradient propagation to reduce overfitting and nullify the learning process of noisy samples. The proofs for all propositions and theorems can be found in the appendix.

**Proposition 3.1.** *let the double softmax cross-entropy loss for a paticular sample be* $\mathcal{L}_{ours} = -\log(q_{\tilde{y}})$*, where* $q = softmax(p)$*,* $p = softmax(z)$*, and* $z$ *is the VLM's output logits. Then, the gradient of the loss with respect to* $z$ *is:*

$$\frac{\partial \mathcal{L}_{ours}}{\partial z_i} = p_i \left[ (q_i - \delta_{\tilde{y}i}) + \left(p_{\tilde{y}} - \sum_j p_j q_j\right) \right]$$

The First term $p_i$ in the above equation restricts the absolute value of the overall gradients across all logit terms to be not greater than 1, suppressing the general prompt updates to achieve conservative adaptation for VLM prompts. The first term in the square brackets: $(q_i - \delta_{yi})$ is the softened alignment signal, which enables the model to learn from supervision information with smoother curves. The second term: $(p_y - \sum_j p_j q_j)$ acts as a balancer for the confidence of the model that measures the difference between the on the noisy label and the weighted average confidence across all classes.

Further studies on the double-softmax cross-entropy gradient in Theorem 3.1 reveal its gradient nullification mechanism under mislabeled data for high-confidence prediction.

**Theorem 3.2.** *For high confidence single-softmax prediction inconsistent with noisy label $y$, i.e., $\exists \hat{y} \neq \tilde{y}, p_{\hat{y}} \to 1$, its total gradient quantity under double-softmax cross-entropy loss with respect to logit $z$ will converge to 0:*

$$\lim_{\boldsymbol{p}_{\hat{y}} \to 1, \hat{y} \neq \tilde{y}} \sum_i \left| \frac{\partial \mathcal{L}_{ours}}{\partial z_i} \right| = 0$$

Theorem 3.2 reveals that the double-softmax cross-entropy loss can effectively suppress the gradient from samples with predictions that are highly confident, but discordant with the noisy label. For VLMs with strong prior knowledge, these excluded samples are likely to be mislabeled, which would have harmed the prompt tuning process. As double-softmax cross-entropy shares the same property of producing small gradients for prediction highly accordant with noisy labels as in Equation (4), our method encourages the VLM prompts to learn from ambiguous samples that receive low confidence predictions from the model, magnifying the memorization of novel samples absent in the prior knowledge. By this means, our double-softmax cross-entropy achieves selective gradient suppression self-adaptively.

**Proposition 3.3.** *In a $C$ class classification problem, for any classification model $f$, its double-softmax cross-entropy loss defined in Equation (3) is bounded by:*

$$\log(1 + \frac{C-1}{e}) \leq \mathcal{L}_{ours} \leq \log(e + C - 1)$$

Theorem 3.3 proves that double-softmax cross-entropy shares the similar property of loss boundedness of other denoising methods, such as LogitClip (Wei et al., 2023), which limits the influence of extremely hard or mislabeled samples, suppressing the tendency of overfitting.

Based on Theorem 3.3, we further analyze the robustness of the double-softmax loss in terms of its generalization risk between clean and noisy distributions. Let $f$ be the classifier to be optimized by the loss function $\mathcal{L}$, let $\mathcal{R}_{\mathcal{L}}(f) = \mathbb{E}_{(\boldsymbol{x},y)} \sim \mathcal{P}_{clean}(\mathcal{L}(f(\boldsymbol{x},y))$ be the expected risk of $f$ under clean data, $\mathcal{R}_{\mathcal{L}}^T(f) = \mathbb{E}_{(\boldsymbol{x},\tilde{y})} \sim \mathcal{P}_{noisy}(\mathcal{L}(f(\boldsymbol{x},\tilde{y}))$ be the expected risk of $f$ under instance-independent label noise with transition matrix $T$, and $f^\star$ and $\tilde{f}^\star$ denote the global minimizer of $\mathcal{R}_{\mathcal{L}}(f)$ and $\mathcal{R}_{\mathcal{L}}^T(f)$ respectively.

**Theorem 3.4.** *For any symmetric label noise with noise rate $\eta < 1 - \frac{1}{C}$, the expected risk difference in the clean data distribution between the clean and noisy global optimizer is bounded by :*

$$0 \leq \mathcal{R}_{\mathcal{L}_{ours}}(\tilde{f}^\star) - \mathcal{R}_{\mathcal{L}_{ours}}(f^\star) \leq \log(\frac{e+C-1}{1+e^{-1}(C-1)})M_\eta$$

*where $M_\eta = \frac{\eta}{1-\eta}$.*

**Theorem 3.5.** *Under asymmetric noise with $T_{jk} \leq T_{kk}, \forall j \neq k$, the expected risk in noisy data distribution of clean and noisy global optimizer is bounded by :*

$$0 \leq \mathcal{R}_{\mathcal{L}_{ours}}^T(f^\star) - \mathcal{R}_{\mathcal{L}_{ours}}^T(\tilde{f}^\star) \leq C \log(\frac{e+C-1}{1+e^{-1}(C-1)})P_T$$

*where $P_T = \mathbb{E}_{(\boldsymbol{x},y)} \sim \mathcal{P}_{clean}(T_{kk})$ is a constant that depends on noise pattern.*

The theorems above bound the discrepancy between the optimal classifiers under clean and noisy labels, thereby ensuring consistent performance across these settings.

## 4. Experiments

### 4.1. Experimental Setup

**Dataset and Settings**. To evaluate the performance of our method, we conduct extensive experiments on several datasets, including Caltech101 (Fei-Fei et al., 2004), StanfordCars(Krause et al., 2013), OxfordPets(Parkhi et al., 2012), Flowers102(Nilsback & Zisserman, 2008), Food101(Bossard et al., 2014), FGVCAircraft(Maji et al., 2013), DTD(Cimpoi et al., 2014), EuroSAT(Helber et al., 2019), and UCF101(Soomro et al., 2012). These datasets cover classification tasks across generic objects, fine-grained objects, scenes, textures, and actions. We use the same dataset splitting strategy as in CoOp(Zhou et al., 2022b). In our experiments, we introduce two different types of noise as mentioned in the preliminary, with noise rates set to {40%, 60%, 80%} for symmetric noise and { 20%, 30%, 40%} for pair-flip noise to simulate moderate and heavy noise scenarios.

**Baselines** We compare our method with multiple baselines, each adopting distinct noise-robust learning strategies. These baselines include:

- Zero-Shot: Using CLIP's zero-shot prediction directly without prompt-tuning.

- CoOp(Zhou et al., 2022b): A basic prompt-tuning method for vision-language models inherently robust to label noise.

- LogitNorm(Wei et al., 2022): A learning method that bounds the magnitude of the model's logit output via $\ell_2$-normalization with an adjustable hyper-parameter.

- Smoothing(Lukasik et al., 2020): A classic noisy label learning tactic which mixes the one-hot ground-truth label with a uniform distribution for regularization.

- NLPrompt(Pan et al., 2025): A specialized noisy label learning algorithm for prompt-tuning that generates pseudo-label via optimal transport for clean sample selection, and introduces MAE loss (Ghosh et al., 2017a) for wrongly labeled samples.

*Table 1.* Final accuracy (%) of tested methods on different datasets and various noise conditions. The bolded number indicates the performance of the best model, and the underlined number indicates the second-best model. All results are the averaged accuracy in the last five epochs and AVG denotes the average accuracy among nine datasets.

| METHOD | DATASET | NOISE TYPE: SYM | | | NOISE TYPE: PAIR | | | DATASET | NOISE TYPE: SYM | | | NOISE TYPE: PAIR | | |
|---|---|---|---|---|---|---|---|---|---|---|---|---|---|---|
| | | 40% | 60% | 80% | 20% | 30% | 40% | | 40% | 60% | 80% | 20% | 30% | 40% |
| ZEROSHOT | CALTECH 101 | 81.43 | | | | | | FGVC AIRCRAFT | 24.37 | | | | | |
| COOP | | 91.06 | 89.37 | 80.00 | 86.14 | 76.38 | 63.21 | | 34.67 | 25.84 | 12.27 | 37.83 | 34.03 | 29.72 |
| SMOOTHING | | 92.82 | 90.54 | 81.29 | 90.39 | 83.77 | 70.47 | | 33.33 | 26.02 | 11.80 | 36.98 | 33.27 | 29.28 |
| LOGITNORM | | 83.06 | 82.14 | 80.21 | 86.44 | 85.27 | 82.71 | | 26.73 | 23.07 | 17.56 | 29.01 | 26.41 | 25.26 |
| NLPROMPT | | **95.36** | 94.42 | 91.85 | 95.56 | 95.21 | 93.63 | | 34.74 | 33.38 | 14.86 | 36.71 | 34.45 | 33.60 |
| DSPT | | 95.07 | **95.31** | **94.01** | **96.06** | **95.85** | **93.71** | | **36.46** | **33.53** | **29.12** | **38.46** | **35.93** | **33.60** |
| ZEROSHOT | STANFORD CARS | 65.33 | | | | | | DTD | 42.73 | | | | | |
| COOP | | 71.56 | 61.76 | 40.48 | 69.38 | 59.99 | 49.47 | | 64.55 | 56.73 | 36.99 | 64.73 | 56.69 | 47.53 |
| SMOOTHING | | 70.58 | 61.45 | 40.90 | 71.61 | 65.21 | 52.51 | | 67.77 | 57.15 | 39.95 | 68.60 | 59.98 | 49.94 |
| LOGITNORM | | 57.05 | 56.97 | 54.05 | 57.96 | 56.49 | 51.48 | | 66.84 | 61.81 | 49.07 | 69.43 | 66.50 | 59.30 |
| NLPROMPT | | **79.70** | **77.21** | **71.98** | 80.90 | 78.48 | 75.01 | | 65.97 | 59.17 | 41.25 | 70.17 | **70.07** | 65.70 |
| DSPT | | 79.64 | 76.78 | 71.50 | **81.14** | **78.82** | **75.15** | | **68.84** | **63.85** | **55.50** | **71.03** | 69.93 | **66.74** |
| ZEROSHOT | OXFORD PETS | 88.19 | | | | | | EUROSAT | 42.91 | | | | | |
| COOP | | 89.17 | 83.05 | 60.43 | 83.22 | 72.96 | 60.76 | | 93.09 | 90.75 | 76.03 | 89.75 | 82.43 | 69.77 |
| SMOOTHING | | 89.38 | 85.85 | 62.76 | 87.97 | 79.41 | 65.58 | | 92.03 | 91.00 | 77.41 | 92.56 | 86.99 | 73.65 |
| LOGITNORM | | 90.75 | 84.18 | 79.02 | 89.73 | 83.46 | 72.27 | | 92.30 | 91.33 | 81.00 | 92.86 | 90.91 | 83.31 |
| NLPROMPT | | 92.41 | 88.97 | 84.18 | 93.63 | 92.85 | 91.27 | | 92.04 | 88.83 | 72.01 | 93.93 | 93.19 | 82.00 |
| DSPT | | **93.00** | **92.60** | **91.25** | 93.19 | 92.83 | **91.74** | | **93.39** | **92.64** | **82.06** | **94.42** | **93.81** | **92.38** |
| ZEROSHOT | FLOWERS 102 | 66.19 | | | | | | UCF101 | 65.19 | | | | | |
| COOP | | 91.03 | 84.54 | 66.40 | 86.27 | 74.99 | 59.39 | | 79.88 | 75.33 | 62.89 | 73.20 | 65.53 | 54.78 |
| SMOOTHING | | 91.89 | 83.89 | 66.91 | **91.09** | 81.44 | 65.28 | | 79.97 | 76.31 | 63.57 | 78.99 | 72.26 | 59.06 |
| LOGITNORM | | 73.11 | 68.96 | 60.82 | 76.01 | 72.89 | 65.15 | | 75.73 | 72.43 | 64.89 | 76.31 | 73.90 | 69.30 |
| NLPROMPT | | **93.29** | 84.80 | 80.08 | 90.37 | **91.47** | 88.39 | | 83.00 | **82.04** | **77.40** | 83.75 | 82.72 | **81.55** |
| DSPT | | 91.12 | **88.15** | **80.35** | 91.01 | 86.93 | **90.07** | | **83.56** | 81.19 | 76.48 | 83.37 | **83.01** | 81.13 |
| ZEROSHOT | FOOD101 | 85.46 | | | | | | AVG | 62.50 | | | | | |
| COOP | | 87.76 | 86.63 | 84.45 | 79.58 | 70.71 | 58.60 | | 78.09 | 72.67 | 57.77 | 74.46 | 65.97 | 54.80 |
| SMOOTHING | | 87.71 | 86.76 | 84.66 | 84.01 | 77.77 | 64.24 | | 78.39 | 73.22 | 58.81 | 78.02 | 71.12 | 58.89 |
| LOGITNORM | | 85.32 | 85.61 | 84.73 | 84.71 | 83.86 | 79.05 | | 72.32 | 69.61 | 63.49 | 73.60 | 71.07 | 65.31 |
| NLPROMPT | | 88.50 | 88.08 | 85.78 | **89.05** | 88.65 | 88.41 | | 80.56 | 77.43 | 68.82 | 81.56 | **80.79** | 77.73 |
| DSPT | | **88.86** | **88.62** | **87.73** | 89.03 | **88.96** | **88.77** | | **81.10** | **79.19** | **74.22** | **81.97** | 80.67 | **79.25** |

**Implementation Details** In our experiment, we use pre-trained CLIP (Radford et al., 2021) as the backbone model, with the ViT-B/16 (Dosovitskiy et al., 2021) as the image encoder. Following the same setting as CoOp(Zhou et al., 2022b), we introduce a randomly initialized prompt of length 16 as learnable parameters shared across all classes while keeping the entire vision and text encoder frozen through the training process. The total number of training epochs is set to 50 for all datasets. The starting learning rate is set to 0.002 with cosine annealing, and the final learning rate decreases to zero. The hyperparameter is set to 0.2 for label smoothing and 1 for LogitNorm. To ensure experimental stability, the reported results are the average test accuracy of the last five epochs. All experiments are implemented by PyTorch (Paszke et al., 2019) and conducted on NVIDIA A800 80GB GPU. Notably, our experiments are conducted on the entire training set with the batch size of 32; this differs from previous prompt-tuning approaches for noisy label learning, which use 16-shot few-shot training.

## 4.2. Experimental Results

The results of our comparative studies are presented in Table 1, and the AVG column indicates the average accuracy of a specific setting among nine datasets. Our method achieves the best performance in most cases, while achieving the second-best performance and being compatible with the NLPrompt in the remaining cases, with an accuracy difference of less than 1% except on Flowers102. Our method also has the best averaged accuracy, which is 0.4% to more than 5% higher than NLPrompt, except on 30% pair-flip noise, in which the difference in accuracy is nearly negligible. Experiments on more noise settings on Caltech101 and OxfordPets are shown in Figure 1.

Beyond the overall performance gain of our method, it can also be derived from Table 1 that CoOp demonstrates a degree of noise robustness as it maintains high accuracy under moderate noise levels on simple datasets such as Catltech 101, in which it still acquires 80% accuracy in 80% sym noise. However, as the noise level increases, CoOp experiences a significant degradation, with an accuracy drop of

*Table 2.* Final accuracy (%) of different methods on Caltech101 and DTD datasets with extremely heavy symmetric and pair-flip noise. The bolded number indicates the performance of the best model, and the underlined number indicates the second-best model.

| DATASET | NOISE TYPE | COOP | SMOOTHING | BOOTSTRAP | LOGITNORM | SELECT | NLPROMT | DSPT |
|---|---|---|---|---|---|---|---|---|
| CALTECH101 | SYM 90% | 53.75 | 61.36 | 58.10 | 78.67 | 78.89 | **87.80** | 86.68 |
| | PAIR 80% | 10.53 | 5.94 | 9.41 | 4.60 | 12.76 | 80.32 | **80.73** |
| DTD | SYM 90% | 21.19 | 19.75 | 22.36 | 29.08 | 29.91 | 36.60 | **40.02** |
| | PAIR 80% | 8.58 | 5.78 | 6.99 | 1.48 | 5.44 | 30.93 | **33.66** |

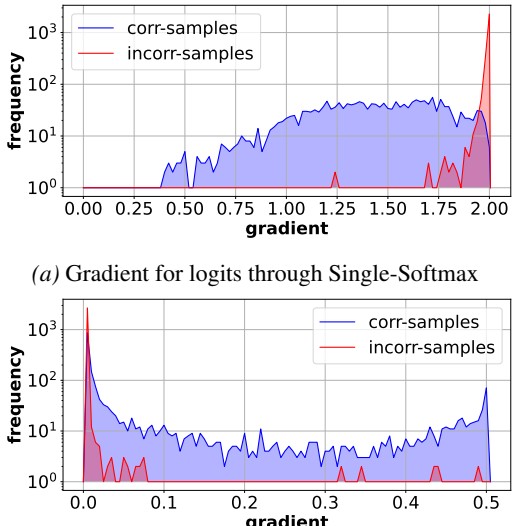

*(a)* Gradient for logits through Single-Softmax

*(b)* Gradient for logits through Double-Softmax

*Figure 3.* Overall gradient propagated to logit $z$ of each correctly labeled and mislabeled sample loss in the first training epoch without parameter update on Caltech101.

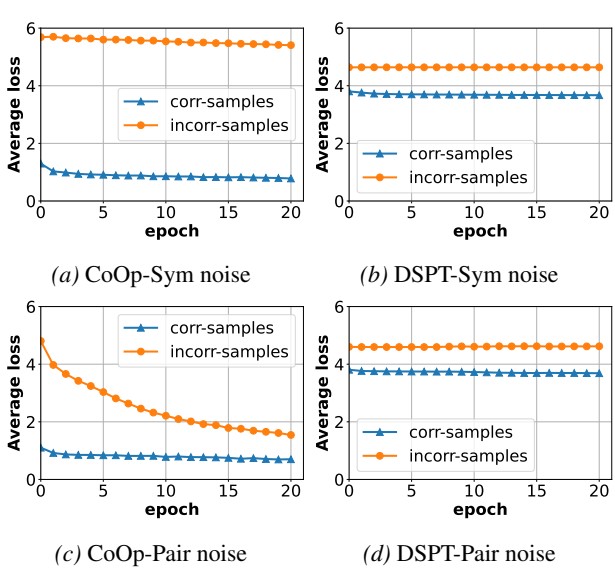

*(a)* CoOp-Sym noise   *(b)* DSPT-Sym noise

*(c)* CoOp-Pair noise   *(d)* DSPT-Pair noise

*Figure 4.* Loss curves of correctly labeled and mislabeled samples in the training process on Caltech101.

more than 20% when the symmetric noise rate rises from 40% to 80 % in StanfordCars and FGVCAircraft. In contrast, LogitNorm performs poorly under low noise settings, but receives relatively high averaged accuracy under 80% symmetric noise and 40% pair noise, which is more than 5% higher than Label Smoothing. Meanwhile, the sample-selection-based method NLPrompt exhibits strong noise robustness under both small and large noise rates.

### 4.3. Further Analysis

*Q1.Does Double-Softmax Cross-Entropy Loss Suppress Gradient Propagation Selectively and Bound the Loss Function?*

*A1.*The gradient suppression and boundedness of double-softmax cross-entropy are examined by experiments. We record the sum of the absolute gradient propagated from each correctly labeled and mislabeled sample loss to logit $z$ in the training process in the first training epoch without parameter updates. This experiment is conducted on Caltech101 under 60% symmetric noise. As shown in Figure 3, the model trained with either single or double softmax cross-entropy loss shows a relatively smooth gradient distribution for samples with correct labels. However, the model with

standard cross-entropy loss has disproportionately large gradients for "incorrect" noisy samples near the maximum value of 2. Conversely, our method not only has a smaller average gradient for clean samples but can also suppress the gradient of mislabeled samples to approximately zero, allowing the correctly labeled samples to dominate the training process. Additional studies in the appendix confirm that this phenomenon is consistent throughout the early stage of the training process.

In addition, we record the average loss among samples with correct and incorrect labels in the first 20 epochs in the training process, with 40% symmetric and pair-flip noise on the Caltech101 dataset. The results can be found in Figure 4. The CoOp model shows large differences between the losses of correctly and incorrectly labeled samples, while both the correct and incorrect average losses of our method stay near the value of 4 persistently, qualifying its robustness.

*Q2.Does Double-Softmax Cross-Entropy Loss Work Under Severe Label Noise?*

*A2.*Our approach still works under high noise ratios. To further analyze the noise robustness of our method, we conduct additional experiments introducing extremely high-level label noise, i.e, 90% sym noise and 80% pair-flip noise on

*Table 3.* Final accuracy (%) of different methods on Caltech101 and OxfordPets dataset with similar loss smoothing and bounding approaches. The bolded number indicates the performance of the best model.

| DATASET | NOISE TYPE | SMOOTHING | LOGITNORM | SQUARE | BOOTSTRAP | NCE | LOGITCLIP | DSPT |
|---|---|---|---|---|---|---|---|---|
| CALTECH101 | SYM 60% | 90.54 | 82.14 | 93.69 | 92.55 | 91.18 | 49.31 | **95.31** |
| | PAIR 30% | 83.77 | 85.27 | 81.59 | 77.80 | 78.31 | 47.47 | **95.85** |
| OXFORDPETS | SYM 60% | 85.85 | 84.18 | 87.52 | 84.69 | 77.05 | 84.96 | **92.60** |
| | PAIR 30% | 79.41 | 83.46 | 75.94 | 74.96 | 71.91 | 75.02 | **92.83** |

Caltech101 and DTD datasets. The baselines include several noisy label learning approaches and the naive sample-selection strategy, in which the inconsistent samples with $argmax_i(\boldsymbol{z}_i) \neq \tilde{y}$ are excluded in the current epoch. Note that the noise becomes dominant for 80% pair-flip noise, creating a challenging task for prompt-tuning. As shown in Table 2, the prompt-tuning method CoOp, label smoothing, and Logit norm are severely dampened, especially under 80% pair noise, in which the accuracy of these three methods drops to lower than 12%. Though still being interfered by label noise, our method demonstrates high performance, outperforming NLPrompt by 0.5%-3% in most cases, and only about 1% lower than NLPrompt in Caltech101 with symmetric noise. In addition, our method shows a significant advantage compared to naive selection, confirming that our method is not totally equivalent to simply picking out consistant sample.

*Q3.Can Similar NLL Methods Achieve Compatible Performance for Prompt-Tuning?*

*A3.*Our method shows advantages compared with similar designs in prompt-tuning with noisy labels. Numerous related studies also introduce bounded loss or similar loss-based smoothing and normalization strategies for NLL. Therefore, we introduce comparative studies involving several related approaches. Apart from Label Smoothing and LogitNorm, which have already been introduced in the previous experiments, this experiment also includes: NCE (Ma et al., 2020), the normalized cross entropy loss, which divides the cross entropy loss by the sum of losses under all classes to fit the symmetric loss pattern. Bootstrapping (Reed et al., 2015), which mixes the model's own prediction into the ground truth label to smooth the training process. Additionally, we introduce a square normalization approach in which the logit is normalized before a quadratic multiplication. This experiment is conducted on Caltech101 and Oxfordpets datasets, with 60% symmetric noise and 30% pair flip noise. The outcome of this experiment is presented in Table 3, in which our model shows its superiority by outperforming all other methods by approximately 2% on Caltech101 with symmetric noise and more than 5% in all other cases.

*Q4.Why Is LogitClip Not Tested in The Main Experiment?*

*A4.*LogitClip is too sensitive to hyperparameters to be tested. LogitClip (Wei et al., 2023) normalizes output logits whose $\ell_2$-norm exceeds a predefined threshold $\tau$, which serves as

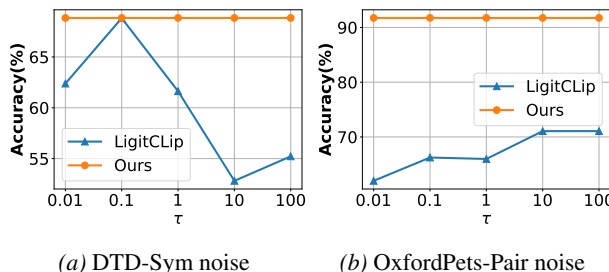

*(a)* DTD-Sym noise    *(b)* OxfordPets-Pair noise

*Figure 5.* Accuracy of LogitClip affected by $\tau$ compared to ours.

a hyperparameter. In our CLIP prompt-tuning scenario, we observe that although LogitClip shares similar strategies and characteristics with our approach, its performance is highly sensitive to the choice of this hyperparameter. Furthermore, the optimal threshold value varies across different datasets and noise settings, making it difficult to adapt LogitClip consistently in diverse scenarios. As shown in Figure 5, we present the accuracy of LogitClip and our method on DTD with 40% symmetric noise and OxfordPets with 40% pair noise. The optimal threshold in the first setting is approximately 0.1, whereas it exceeds 1 in the second setting. In Figure 5a, the change of $\tau$ causes accuracy drop of more than 15%. These results demonstrate that our method achieves not only superior performance but also greater flexibility compared to LogitClip.

## 5. Conclusion

In this paper, we introduced Double-Softmax Prompt Tuning (DSPT), a simple yet effective framework for vision-language models in prompt tuning with label noise. By leveraging additional normalization, DSPT transforms the traditionally problematic phenomenon of "gradient vanishing" into a principled noise-filtering shield. Our method induces a self-adaptive saturation zone that effectively "zeros out" disruptive gradients from high-error noisy samples while selectively permitting informative updates for reliable adaptation. Our extensive experimental results across various benchmarks demonstrate that DSPT achieves state-of-the-art robustness. Theoretical analysis further confirms that the double-softmax cross-entropy loss provides intrinsic gradient suppression and ensures loss boundedness, preventing the model from fitting incorrect labels. Looking ahead, our future work is to extend double-softmax prompt tuning beyond CLIP-based classification to large vision–language

models (LVLM), exploring its effectiveness for LVLM-based downstream noisy label learning. We believe that the simplicity, generality, and strong empirical robustness of double-softmax make it a promising building block for future noisy label learning frameworks for finetuning multimodal foundation models.

## Impact Statement

This paper presents work in the field of Machine Learning. There are many potential societal consequences of our work, none of which we feel must be highlighted here.

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

# A. Proofs for Theoretical analysis

In this section, we provide detailed proofs for Propositions and Theorems in our paper.

**Proof for Proposition 3.1**

*Proof.*

$$
\begin{aligned}
\frac{\partial \mathcal{L}_{ours}}{\partial \boldsymbol{z}_i} &= \sum_j \frac{\partial \mathcal{L}_{ours}}{\partial \boldsymbol{q}_j} \left( \sum_k \frac{\partial \boldsymbol{q}_j}{\partial \boldsymbol{p}_k} \cdot \frac{\partial \boldsymbol{p}_k}{\partial \boldsymbol{z}_i} \right) \\
&= -\frac{1}{\boldsymbol{q}_y} \sum_k \frac{\partial \boldsymbol{q}_{\tilde{y}}}{\partial \boldsymbol{p}_k} \cdot \frac{\partial \boldsymbol{p}_k}{\partial \boldsymbol{z}_i}
\end{aligned}
$$

For $\boldsymbol{s} = softmax(\boldsymbol{a})$, the derivative is $\frac{\partial \boldsymbol{s}_m}{\partial \boldsymbol{a}_n} = \boldsymbol{s}_m(\delta_{mn} - \boldsymbol{s}_n)$. Therefore:

$$
\begin{aligned}
\frac{\partial \mathcal{L}_{ours}}{\partial \boldsymbol{z}_i} &= -\frac{1}{\boldsymbol{q}_{\tilde{y}}} \sum_k (\boldsymbol{q}_{\tilde{y}}(\delta_{\tilde{y}k} - \boldsymbol{q}_k)) \cdot (\boldsymbol{p}_k(\delta_{ki} - \boldsymbol{p}_i)) \\
&= -\sum_k (\delta_{\tilde{y}k} - \boldsymbol{q}_k) \cdot (\boldsymbol{p}_k(\delta_{ki} - \boldsymbol{p}_i)) \\
&= -\sum_k \delta_{\tilde{y}k} \boldsymbol{p}_k(\delta_{ki} - \boldsymbol{p}_i) - \boldsymbol{q}_k \boldsymbol{p}_k(\delta_{ki} - \boldsymbol{p}_i) \\
&= -(\boldsymbol{p}_{\tilde{y}}(\delta_{\tilde{y}i} - \boldsymbol{p}_i) - \boldsymbol{p}_i(\boldsymbol{q}_i - \sum_k \boldsymbol{p}_k \boldsymbol{q}_k)) \\
&= \boldsymbol{p}_i \boldsymbol{q}_i - \boldsymbol{p}_i \sum_k \boldsymbol{p}_k \boldsymbol{q}_k - \boldsymbol{p}_{\tilde{y}} \delta_{\tilde{y}i} + \boldsymbol{p}_{\tilde{y}} \boldsymbol{p}_i
\end{aligned}
$$

Because $\forall i, \boldsymbol{p}_{\tilde{y}} \delta_{\tilde{y}i} = \boldsymbol{p}_i \delta_{\tilde{y}i}$:

$$
\begin{aligned}
\frac{\partial \mathcal{L}_{ours}}{\partial \boldsymbol{z}_i} &= \boldsymbol{p}_i \boldsymbol{q}_i - \boldsymbol{p}_i \sum_k \boldsymbol{p}_k \boldsymbol{q}_k - \boldsymbol{p}_i \delta_{\tilde{y}i} + \boldsymbol{p}_i \boldsymbol{p}_{\tilde{y}} \\
&= \boldsymbol{p}_i \left[ (\boldsymbol{q}_i - \delta_{\tilde{y}i}) + (\boldsymbol{p}_{\tilde{y}} - \sum_j \boldsymbol{p}_j \boldsymbol{q}_j) \right]
\end{aligned}
$$

Which concludes the proof.

**Proof for Theorem 3.2**

*Proof.* Let $\boldsymbol{p}_{\hat{y}} = 1 - \delta$ where $\delta > 0$, then:

For $\forall i \neq \hat{y}$:

$$
\begin{aligned}
|\frac{\partial \mathcal{L}_{ours}}{\partial \boldsymbol{z}_i}| &= \left| \boldsymbol{p}_i \left[ (\boldsymbol{q}_i - \delta_{\tilde{y}i}) + (\boldsymbol{p}_{\tilde{y}} - \sum_j \boldsymbol{p}_j \boldsymbol{q}_j) \right] \right| \\
&\leq 2|\boldsymbol{p}_i|
\end{aligned}
$$

Therefore:

$$
\sum_{i \neq \hat{y}} \left| \frac{\partial \mathcal{L}_{ours}}{\partial \boldsymbol{z}_i} \right| \leq \sum_{i \neq \hat{y}} |\boldsymbol{p}_i| = 2\delta
$$

For $i = \hat{y}$:

$$\left|\frac{\partial \mathcal{L}_{ours}}{\partial \boldsymbol{z}_i}\right| = \left|\frac{\partial \mathcal{L}_{ours}}{\partial \boldsymbol{z}_{\hat{y}}}\right|$$

$$= \left|\boldsymbol{p}_{\hat{y}}\left[\boldsymbol{q}_{\hat{y}} + \left(\boldsymbol{p}_{\tilde{y}} - \sum_j \boldsymbol{p}_j \boldsymbol{q}_j\right)\right]\right|$$

$$\leq \left|\left[\boldsymbol{q}_{\hat{y}} + \left(\boldsymbol{p}_{\tilde{y}} - \sum_j \boldsymbol{p}_j \boldsymbol{q}_j\right)\right]\right|$$

$$= \left|\delta \boldsymbol{q}_{\hat{y}} + \boldsymbol{p}_{\tilde{y}} - \sum_{j \neq \hat{y}} \boldsymbol{p}_j \boldsymbol{q}_j\right|$$

$$\leq \delta \boldsymbol{q}_{\hat{y}} + \boldsymbol{p}_{\tilde{y}} + \sum_{j \neq \hat{y}} \boldsymbol{p}_j \boldsymbol{q}_j$$

$$\leq \delta \boldsymbol{q}_{\hat{y}} + \boldsymbol{p}_{\tilde{y}} + \sum_{j \neq \hat{y}} \boldsymbol{p}_j$$

$$= \delta \boldsymbol{q}_{\hat{y}} + \boldsymbol{p}_{\tilde{y}} + \delta$$

$$\leq 3\delta$$

Therefore:

$$\sum_i \left|\frac{\partial \mathcal{L}_{ours}}{\partial \boldsymbol{z}_i}\right| \leq 5\delta$$

Then, for all $\epsilon > 0$, there always exists $0 < \delta < \frac{\epsilon}{5}$ such that:

$$\sum_i \left|\frac{\partial \mathcal{L}_{ours}}{\partial \boldsymbol{z}_i}\right| \leq \epsilon$$

which concludes the proof.

**Proof for Proposition 3.3**

*Proof.* For any softmaxed vector $\boldsymbol{q}$, its exponential $\ell_1$-norm reaches the maximum when it is a one-hot vector, where $\exists i, \boldsymbol{q}_i = 1$ and $\forall j \neq i, \boldsymbol{q}_i = 0$. In this case:

$$\|e^{\boldsymbol{q}}\|_1 = e + C - 1$$

where $C$ is the length of $\boldsymbol{q}$.

and because the maximum and minimum value for $\forall i, \boldsymbol{q}_i$, is 1 and 0, which are both accessible when $\boldsymbol{q}$ is a one-hot vector, therefore:

$$\frac{1}{e + C - 1} \leq \boldsymbol{p}_i \leq \frac{e}{e + C - 1}, \forall i$$

where $\boldsymbol{p} = softmax(\boldsymbol{q})$.

Consequently, we have that:

$$\log\left(1 + \frac{C - 1}{e}\right) \leq \mathcal{L}_{\text{ours}} \leq \log(e + C - 1)$$

which concludes the proof.

**Proof for Theorem 3.4**

*Proof.* For any classifier model $f$, its expected risk under symmetric label noise and double-softmax cross-entropy loss can be rewritten as:

$$
\begin{aligned}
\mathcal{R}^T{}_{\mathcal{L}_{\text{ours}}}(f) \\
&= \mathbb{E}_{(\boldsymbol{x},\tilde{y}) \sim \mathcal{P}_{noisy}}[\mathcal{L}_{\text{ours}}(f(\boldsymbol{x},\tilde{y})] \\
&= \mathbb{E}_{(\boldsymbol{x},y) \sim \mathcal{P}_{clean}}[\mathbb{E}_{\tilde{y}|y}[\mathcal{L}_{\text{ours}}(f(\boldsymbol{x},\tilde{y}))]] \\
&= \mathbb{E}_{(\boldsymbol{x},y) \sim \mathcal{P}_{clean}}[(1-\eta)\mathcal{L}_{\text{ours}}(f(\boldsymbol{x},y)) + \sum_{i \neq y} \frac{\eta}{C-1}\mathcal{L}_{\text{ours}}(f(\boldsymbol{x},i))] \\
&= (1-\eta)\mathcal{R}_{\mathcal{L}_{\text{ours}}}(f) + \mathbb{E}_{(\boldsymbol{x},y) \sim \mathcal{P}_{clean}}[\sum_{i \neq y} \frac{\eta}{C-1}\mathcal{L}_{\text{ours}}(f(\boldsymbol{x},i))]
\end{aligned}
$$

From Theorem 3.3, we have that:

$$
\log(1 + \frac{C-1}{e}) \leq \mathcal{L}_{\text{ours}} \leq \log(e + C - 1)
$$

Therefore:

$$
(1-\eta)\mathcal{R}_{\mathcal{L}_{\text{ours}}}(f) + \eta\log(1 + \frac{C-1}{e}) \leq \mathcal{R}^T_{\mathcal{L}_{\text{ours}}}(f) \leq (1-\eta)\mathcal{R}_{\mathcal{L}_{\text{ours}}}(f) + \eta\log(e + C - 1)
$$

This inequality can be reformed as:

$$
\frac{1}{1-\eta}(\mathcal{R}^T_{\mathcal{L}_{\text{ours}}}(f) - \eta\log(e + C - 1)) \leq \mathcal{R}_{\mathcal{L}_{\text{ours}}}(f) \leq \frac{1}{1-\eta}(\mathcal{R}^T_{\mathcal{L}_{\text{ours}}}(f) - \eta\log(1 + \frac{C-1}{e}))
$$

Thus:

$$
\mathcal{R}_{\mathcal{L}_{\text{ours}}}(\tilde{f}^\star) - \mathcal{R}_{\mathcal{L}_{\text{ours}}}(f^\star) \leq \frac{1}{1-\eta}(\mathcal{R}^T_{\mathcal{L}_{\text{ours}}}(\tilde{f}^\star) - \mathcal{R}^T_{\mathcal{L}_{\text{ours}}}(f^\star) + \eta\log(\frac{e+C-1}{1+e^{-1}(C-1)}))
$$

Since $\mathcal{R}_{\mathcal{L}_{\text{ours}}}(\tilde{f}^\star) - \mathcal{R}_{\mathcal{L}_{\text{ours}}}(f^\star) \geq 0$ and $\mathcal{R}^T_{\mathcal{L}_{\text{ours}}}(\tilde{f}^\star) - \mathcal{R}^T_{\mathcal{L}_{\text{ours}}}(f^\star) \leq 0$ as $f^\star$ and $\tilde{f}^\star$ are the global minimizer of $\mathcal{R}_{\mathcal{L}_{\text{ours}}}(f)$ and $\mathcal{R}^T_{\mathcal{L}_{\text{ours}}}(f)$ respectively, we have:

$$
0 \leq \mathcal{R}_{\mathcal{L}_{\text{ours}}}(\tilde{f}^\star) - \mathcal{R}_{\mathcal{L}_{\text{ours}}}(f^\star) \leq \frac{\eta}{1-\eta}\log(\frac{e+C-1}{1+e^{-1}(C-1)})
$$

which concludes the proof.

**Proof for Theorem 3.5**

*Proof.* For any classifier model $f$, its expected risk under asymmetric label noise and double-softmax cross-entropy loss can be rewritten as:

$$
\begin{aligned}
\mathcal{R}^T{}_{\mathcal{L}_{\text{ours}}}(f) \\
&= \mathbb{E}_{(\boldsymbol{x},\tilde{y}) \sim \mathcal{P}_{noisy}}[\mathcal{L}_{\text{ours}}(f(\boldsymbol{x},\tilde{y})] \\
&= \mathbb{E}_{(\boldsymbol{x},y) \sim \mathcal{P}_{clean}}[\mathbb{E}_{\tilde{y}|y}[\mathcal{L}_{\text{ours}}(f(\boldsymbol{x},\tilde{y}))]] \\
&= \mathbb{E}_{(\boldsymbol{x},y) \sim \mathcal{P}_{clean}}[T_{yy}\mathcal{L}_{\text{ours}}(f(\boldsymbol{x},y)) + \sum_{i \neq y} T_{yi}\mathcal{L}_{\text{ours}}(f(\boldsymbol{x},i))] \\
&\leq \mathbb{E}_{(\boldsymbol{x},y) \sim \mathcal{P}_{clean}}[T_{yy}(C\log(e+C-1) - \sum_{i \neq y}\mathcal{L}_{\text{ours}}(f(\boldsymbol{x},i)) + \sum_{i \neq y} T_{yi}\mathcal{L}_{\text{ours}}(f(\boldsymbol{x},i))] \\
&= C\log(1 + e(C-1)) \cdot \mathbb{E}_{(\boldsymbol{x},y) \sim \mathcal{P}_{clean}}[T_{yy}] - \mathbb{E}_{(\boldsymbol{x},y) \sim \mathcal{P}_{clean}}[\sum_{i \neq y}(T_{yy} - T_{yi})\mathcal{L}_{\text{ours}}(f(\boldsymbol{x},i))]
\end{aligned}
$$

On the other hand, we also have that:

$$\mathcal{R}^T{}_{\mathcal{L}_{\text{ours}}}(f) \geq C \log(1 + \frac{C-1}{e}) \cdot \mathbb{E}_{(\boldsymbol{x},y)} \sim \mathcal{P}_{clean}[T_{yy}] - \mathbb{E}_{(\boldsymbol{x},y)} \sim \mathcal{P}_{clean}[\sum_{i \neq y}(T_{yy} - T_{yi})\mathcal{L}_{\text{ours}}(f(\boldsymbol{x},i))]$$

Therefore:

$$\mathcal{R}^T{}_{\mathcal{L}_{\text{ours}}}(f^\star) - \mathcal{R}^T_{\mathcal{L}_{\text{ours}}}(\tilde{f}^\star) \leq$$
$$C \log(\frac{e+C-1}{1+e^{-1}(C-1)}) \cdot \mathbb{E}_{(\boldsymbol{x},y)} \sim \mathcal{P}_{clean}[T_{yy}]$$
$$+ \mathbb{E}_{(\boldsymbol{x},y)} \sim \mathcal{P}_{clean}[\sum_{i \neq y}(T_{yy} - T_{yi})(\mathcal{L}_{\text{ours}}(\tilde{f}^\star(\boldsymbol{x},i))$$
$$- \mathcal{L}_{\text{ours}}(f^\star(\boldsymbol{x},i)))]$$

Let $\boldsymbol{z}'^\star$ be the softmaxed output logit of $f^\star$ for sample $(\boldsymbol{x},y)$. Because $f^\star$ is the global minimizer of clean empirical risk, it is the minimizer of $\mathcal{L}_{\text{ours}}(f(\boldsymbol{x},y))$, where $\boldsymbol{z}'^\star_y = 1$ and $\forall i \neq y, \boldsymbol{z}'^\star_i = 0$, which is also the maximizer of $\forall i \neq y, \mathcal{L}_{\text{ours}}(f(\boldsymbol{x},i))$, and as $T_{yy} - T_{yi} > 0$, consequently:

$$\mathbb{E}_{(\boldsymbol{x},y)} \sim \mathcal{P}_{clean}[\sum_{i \neq y}(T_{yy} - T_{yi})(\mathcal{L}_{\text{ours}}(\tilde{f}^\star(\boldsymbol{x},i)) - \mathcal{L}_{\text{ours}}(f^\star(\boldsymbol{x},i)))] \leq 0$$

Thus:

$$0 \leq \mathcal{R}^T_{\mathcal{L}_{\text{ours}}}(f^\star) - \mathcal{R}^T_{\mathcal{L}_{\text{ours}}}(\tilde{f}^\star) \leq C \log(\frac{e+C-1)}{1+e^{-1}(C-1)}) \cdot \mathbb{E}_{(\boldsymbol{x},y)} \sim \mathcal{P}_{clean}[T_{yy}]$$

which concludes the proof.

## B. Details of Our datasets

In our experiments, we introduce nine datasets, including classification tasks in different domains. These initially reliable datasets are then corrupted by manually adding noise into sample labels. In this section, we provide a detailed introduction to these datasets, including their original tasks, their class numbers, features of their samples, and the size of their training and testing sets.

- Caltech101: An object recognition dataset comprising 101 different types of generic categories, with each photo's resolution being roughly $300 \times 200$. The size of the training set is 4128, and the size of the training set is 2465.

- StanfordCars: A fine-grained car recognition dataset that includes 196 different kinds of cars, with class labels depicting their brands, models, and years. The training size of StanfordCars is 6509, and the testing size is 8041.

- OxfordPets: A fine-grained pets breeds recognition dataset including pictures of 37 kinds of cats and dogs, with large variations in scale, pose, and lighting. The training size of StanfordCars is 2944, and the testing size is 3669.

- Flowers102: A fine-grained flower recognition dataset consists of 102 kinds of flowers, with the training size of 4093 and testing size of 2463.

- Food101: A large-scale food classification dataset containing food pictures from 101 categories, with a maximum width of 512 pixels. The size of the training set in our experiment is 50500 and the size of the testing set is 30300.

- FGVCAircraft: A fine-grained aircraft classification dataset consisting of 102 types of airplanes with distinct variants, families, and manufacturer names. The 102,100 samples are split evenly for training, validating, and testing. This dataset is challenging due to the large size of the pictures and the difficulty of distinguishing different aircraft models.

*Table 4.* Accuracy (%) of zero-shot prediction and robust unsupervised prompt-tuning method on nine datasets. The AVG indicates the average accuracy among all datasets.

| Dataset | Caltech101 | StanfordCars | Oxfordpets | Flowers102 | Food101 |
|---------|-----------|--------------|------------|------------|---------|
| Zero-shot | 81.43 | 65.33 | 88.19 | 66.99 | 85.46 |
| Robust-UPL | 82.85 | 65.74 | 88.39 | 70.34 | 86.08 |
| Dataset | FGVCAircraft | DTD | EuroSAT | UCF101 | AVG |
| Zero-shot | 24.27 | 42.73 | 42.91 | 65.19 | 62.50 |
| Robust-UPL | 22.47 | 44.93 | 43.08 | 67.07 | 63.43 |

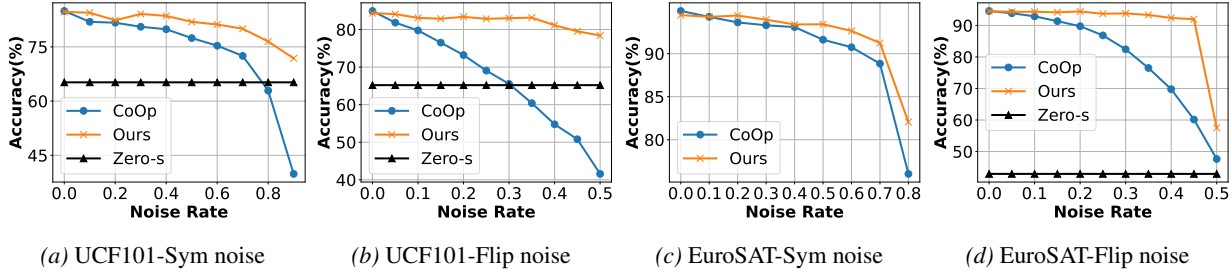

|      (a) UCF101-Sym noise      |      (b) UCF101-Flip noise      |      (c) EuroSAT-Sym noise      |      (d) EuroSAT-Flip noise      |

*Figure 6.* Additional studies on the accuracy curve of prompt-tuning(CoOp), our method, and zero-shot predictions with increasing noise rate under different settings on UCF101 and EuroSAT datasets.

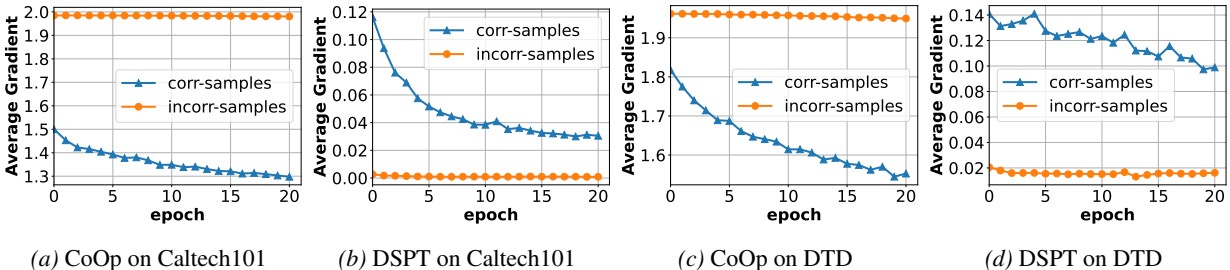

|      (a) CoOp on Caltech101      |      (b) DSPT on Caltech101      |      (c) CoOp on DTD      |      (d) DSPT on DTD      |

*Figure 7.* Studies on the gradient curve of prompt-tuning(CoOp) and our method with respect to CLIP's output logits on Caltech101 and DTD datasets with 60% symmetric noise

- DTD: A texture recognition dataset with 47 classes of textual images. The training size of this dataset is 2820, and the testing size is 1692.

- EuroSAT: A fine-grained satellite recognition dataset containing 10 types of different landscapes, with the size of each picture being $64 \times 64$. The training size of this dataset is 13500, and the testing size is 8100.

- UCF101: A video action recognition dataset including 101 kinds of human movements. The input picture is acquired by cutting midframes from these videos with a resolution of $320 \times 240$. The training set of this dataset is 7639, with testing set of size 3783.

## C. Additional Experiments

### C.1. Experiments on Zero-shot Prediction Methods

Unlike supervised learning methods, which can be heavily degraded by label noise, zero-shot prediction and unsupervised prompt-tuning rely solely on the knowledge learned in the pre-training stage, therefore being unaffected by label noise in the fine-tuning process. In our experiments, we examine the accuracy of zero-shot prediction, which uses the pretrained model for classification directly, and the robust unsupervised prompt-tuning method (Robust-UPL), which treats the zero-shot

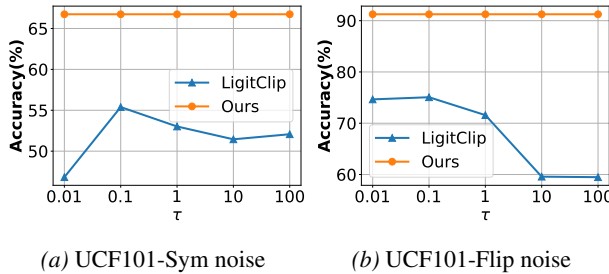

*(a)* UCF101-Sym noise         *(b)* UCF101-Flip noise

*Figure 8.* Additional studies on the effects of hyperparameter $\tau$ of LogitClip compared to ours on UCF101 dataset.

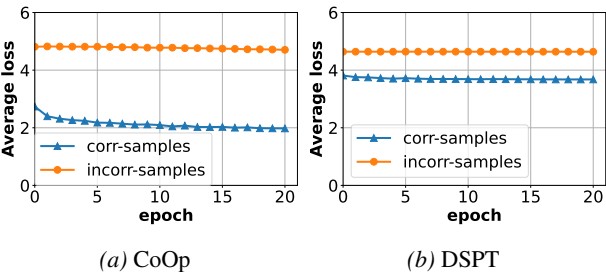

*(a)* CoOp            *(b)* DSPT

*Figure 9.* Additional studies on the accuracy curve of prompt-tuning(CoOp) and our method in the early training stage on Catltech101 dataset with 80% symmetric noise.

*Table 5.* Accuracy (%) on CIFAR10 dataset with symmetric and pair-flip noise. The bolded number indicates the performance of the best model. The NLPrompt method cannot work in this setting.

| Dataset | Noise Type | Noise Rate | CE | LogitClip | Bootstrap | GCE | Smoothing | NLPrompt | DSPT |
|---------|-----------|-----------|-----|-----------|-----------|-----|-----------|----------|------|
| CIFAR10 | Sym | 20% | 74.01 | **84.70** | 74.80 | 84.47 | 76.29 | - | 82.34 |
| | | 50% | 57.79 | 74.56 | 54.86 | 64.77 | 52.34 | - | **77.77** |
| | | 80% | 25.18 | 22.88 | 32.54 | 36.03 | 36.36 | - | **37.93** |
| | Pair | 20% | 81.35 | 82.07 | 80.45 | **83.18** | 81.48 | - | 81.53 |
| | | 40% | 72.14 | **81.02** | 70.31 | 73.27 | 72.09 | - | 57.07 |

predictions as ground-truth labels for prompt tuning. The results in Table 4 show that a pre-trained CLIP model can achieve moderate performance on all datasets, but is still outperformed by robust prompt-tuning methods. Compared with zero-shot prediction, Robust-UPL offers only limited improvement, since it does not introduce additional supervised samples or other useful information. These experiments justify the necessity of using labeled samples, even if the labels are unreliable.

### C.2. Additional Experiments on The effect of Label Noise

In this section, we present further exploration on the learning curves of CoOp, our method, and zero-shot predictions with increasing noise rate on UCF101 and EuroSAT datasets with symmetric and pair-flip label noise. The results can be found in Figure 6. Our method exhibits significant accuracy improvements compared to CoOp, especially under high noise ratios.

### C.3. Experiments on the Gradient Curves in the Training Process

To verify that the double-softmax cross-entropy loss effectiveness of zeroing out the gradient from mislabeled samples, we conduct further experiments that record the average gradient from each correctly and mislabeled propagated to the output logits $z$ on Caltech101 and DTD datasets with 60% symmetric noise in the first 20 training epochs. The results can be found in Figure 7. Our method exhibits continuous noisy gradient suppression compared to CoOp.

### C.4. Additional Experiments on LogitClip's Hyperparameter

To study the effect of hyperparameter $\tau$ on the LogitClip approach, we carried out further experiments on the UCF101 dataset with 80% symmetric noise and 40% flip noise. The results are shown in Figure 8. The accuracy curve is consistent

with our conclusion drawn in Section 4.3, confirming that LogitClip is sensitive to its hyperparameter.

### C.5. Additional Experiments on The Learning Curve of CoOp and Our Method

In this section, we present further exploration of average losses for correct and incorrect predictions in the early training stage, where CoOp and our method are trained on the Caltech101 dataset with 80% symmetric noise. The results are shown in Figure 9.

### C.6. Additional Experiments on Whether Double-Softmax is Suitable for Classical Noisy Label Learning Tasks

To testify double-softmax cross-entropy loss under classical NLL tasks, we conduct experiments on the CIFAR10 (Krizhevsky & Hinton, 2009) image classification datasets with symmetric noise of {20%, 50%, 80%}, and pair-flip noise of {20%, 40%}. We compare our method with other NLL-robust loss functions and mechanisms, including learning with standard cross-entropy loss, Label Smoothing, LogitClip, Bootstrapping (Reed et al., 2015), and Generalized Cross Entropy (GCE) loss (Zhang & Sabuncu, 2018). ResNet34 (He et al., 2016) is used as the backbone, and all models are trained for 200 epochs.

As shown in Tab. 5, our double-softmax method does not perform as well as it does in VLM prompt-tuning. Although our model reaches the highest performance on CIFAR10 with Sym 50% and 80% by the advantage of 1%-3%, it is still outmatched by LogitClip or GCE in other cases, with the accuracy of more than 20% lower than LogitClip on 40% pair-flip noise. This is because double-softmax is specially designed for confident models with strong prior knowledge. It is also worth noticing that NLPrompt (Pan et al., 2025) does not apply to single-modal tasks, as it relies on pseudo-labels generated from text-image feature similarities.

## D. Additional Discussions

*Q1. Why Are The Experiments Conducted on The Full Training Set, Instead of Using Few-Shot Learning?*

*A1.* This work mainly focuses on noisy label learning rather than few-shot learning, and simply mixing up these two settings will introduce unexpected interference. For example, in a 50% symmetric noise scenario, for a dataset with 1000 samples and 10 classes, the number of correctly labeled samples is approximately 50 for each class, while for 4-shot learning, this number can vary from 0 to 4. Previous studies investigating the effect of label noise on few-shot prompt-tuning have to fix the correct number to 2 for stability, which violates the randomness of label noise.

*Q2. Why Is The Double-Softmax Cross-Entropy Loss Unsuitable for Classic Noisy Label Learning?*

*A2.* Double-softmax cross-entropy works by suppressing noisy gradient by model's confident predictions and restricting the loss magnitude. For classical noisy label learning backbone models that are trained from scratch, this mechanism will hinder the learning process in the early training stage. In addition, the values of the logit outputs of the backbone model in traditional NLL are different from those in VLMs, creating over-smoothed output distributions.

*Q3. Can The Double-Softmax Cross-Entropy Be Improved?*

*A3.* As shown in Table 1, although our method has demonstrated state-of-the-art performance on various datasets, it is not always superior to other approaches, especially under moderate label noise cases. To solve this issue, self-adjustive hyperparameters need to be introduced to control the smoothness of the softmaxed logit according to the specific features of the task. In addition, it is also necessary to extend double-softmax prompt-tuning beyond CLIP-based classification to large vision–language models (LVLM), exploring its effectiveness for LVLM-based downstream noisy label learning. We believe that the simplicity, generality, and strong empirical robustness of double-softmax make it a promising building block for future noisy label learning frameworks for finetuning multi-modal foundation models.

