# OpenReview forum: "Intrinsic Gradient Suppression for Label-Noise Prompt Tuning in Vision–Language Models"
_ICML.cc/2026/Conference — ICML 2026 regular_

### Official Review · Reviewer_Yh1U · 2026-03-11

**Soundness:** 3
**Presentation:** 2
**Significance:** 2
**Originality:** 2
**Overall Recommendation:** 4
**Confidence:** 3

**Summary:**

The paper proposes Double-Softmax Prompt Tuning (DSPT), a hyperparameter-free approach for boosting the robustness of prompt tuning for vision-language model (VLM) against noisy labels. The authors observe that the cross-entropy loss causes considerable gradient updates for mislabeled data. Models like CLIP have a strong zero-shot prior and high confidence for the correct class, which is treated as a mismatch for the noisy label. To address the impact of label noise, DSPT uses a double softmax operation on the logits before the cross-entropy loss calculation. This limits the loss, preventing the gradients from being influenced by the high confidence, yet false, predictions. The model's performance is tested on nine image classification benchmarks with symmetric and pair flip label noise. It shows competitive performance against several baselines such as NLPrompt, LogitNorm, and Label Smoothing.

**Compliance With Llm Reviewing Policy:**

Affirmed.

**Final Justification:**

I thank the authors for their rebuttal and the additional experimental work.

I can acknowledge that the discussion has helped to clarify matters and I can consider the majority of the originally raised concerns to be well addressed.

**Key Questions For Authors:**

1. The LogitClip omission from the main paper, due to hyper parameter sensitivity, weakens the empirical claims. What was the performance of the baseline method, default/fixed parameter, in comparison to DSPT?

2. The current evaluation relies solely on CLIP with a ViT-B/16 encoder. Have you evaluated DSPT on larger encoders or different VLM architectures to verify that the gradient surge phenomenon and DSPT's efficacy remain consistent?

3. How does the method distinguish between a genuinely mislabeled sample and a 'hard' out-of-distribution clean sample where the zero-shot prior happens to be wrong?

4. Table 5 demonstrate that DSPT underperforms on CIFAR-10 when trained from scratch. Can DSPT be adapted to bridge this gap for models lacking strong initial priors?

**Limitations:**

Yes, but they are in appendix, not in the main paper.

**Strengths And Weaknesses:**

**Soundness:**

Strength: The empirical evaluation covers a wide range of datasets and noise ratios demonstrating that the method generally prevents catastrophic overfitting. The paper also provides theoretical bounds on the loss function (Proposition 3.3) and generalisation risk (Theorems 3.4 and 3.5).

Weakness: The authors exclude LogitClip from the main comparative experiments (Table 1) claiming it is 'too sensitive to hyperparameters'. A proper evaluation should tune the baseline’s hyperparameter per dataset or noise level or report its performance across a standard range rather than omitting a relevant competing method. The authors claim the method is robust for VLMs but only evaluate a single and small backbone (ViT-B/16). It is unclear if DSPT scales effectively to larger models or different VLM architectures where the logit distributions might behave differently.

**Presentation:**

Strengths: The motivation is clearly articulated and successfully highlights the gradient surge paradox. The transition from theoretical bounds to empirical validation is logical and the paper is easy to follow.

Weakness: Typographical errors like Figure 5 and Figure 8 label the baseline as 'LigitCLip' instead of LogitClip, Table 5 misspells the baseline as 'LogitLcip'. Some of the critical sections (Appendix C.6 and D) should be promoted and discussed in the main paper.

**Significance:**

Strength: The plug-and-play nature of the proposed loss function without any hyper parameter tuning appears effective in certain real-world applications. The method is computationally cheap, easy to implement, and requires no additional modules or explicit sample-selection algorithms.

Weakness: The significance is strictly limited by the necessity of a good pre-trained prior. If the zero-shot performance of the VLM on a target dataset is poor, DSPT will indiscriminately block gradients, thereby preventing the model from learning actual hard examples. It is a rather specialized trick c.f. a fundamental step forward in learning with noisy labels. The authors note in Appendix C.6 that for some methods the introduced approach underperforms baselines.

**Originality:**

Strength:Applying the double-softmax specifically to mitigate gradient surges in VLM prompt-tuning can be considered a somewhat creative intersection of techniques.

Weakness: The algorithmic novelty is marginal. The method essentially acts as a static logit scaling/clipping mechanism, heavily overlapping with the conceptual goals of LogitNorm and LogitClip but without the flexibility of a tunable parameter.

---

> ### Author Rebuttal · Authors · 2026-03-30
>
> Thank you for your time and effort in reviewing our paper.
>
> **W1. & Q1. The authors exclude LogitClip from the main comparative experiments (Table 1)...**
>
> Thank you for your reply. We exclude LogitClip from our main experiment because it suffers from great performance changes under different hyper-parameters as shown in our experiment.
> In addition, changing hyperparameters alone for LogitClip will also introduce unfairness to LogitNorm and Label smoothing, which also require hyperparameters, but exhibit more stable performance than LogitClip. To address the reviewer's concerns, we conduct an experiment
> on DTD and Caltech101 datasets with the same symmetric and pair-flip noise as our main experiment with $\tau = 1$. The result shows that the LogitClip, which was originally designed for classical noisy label learning tasks, is not suitable for our setting.
>
> | DTD-sym     | 40%   | 60%   | 80%   | DTD-flip     | 20%   | 30%   | 40%   |
> |-------------|-------|-------|-------|--------------|-------|-------|-------|
> | LogitClip   | 55.52 | 53.01 | 44.88 | LogitClip    | 56.82 | 57.53 | 52.67 |
> | DSPT        | 68.86 | 63.85 | 55.5  | DSPT         | 71.03 | 69.93 | 66.74 |
> | Caltech-sym | 40%   | 60%   | 80%   | Caltech-flip | 20%   | 30%   | 40%   |
> | LogitClip   | 38.62 | 58.93 | 38.43 | LogitClip    | 45.02 | 47.47 | 41.53 |
> | DSPT        | 95.07 | 95.31 | 94.01 | DSPT         | 96.06 | 95.85 | 93.71 |
>
> **W2. Typographical errors like Figure 5 and Figure 8 label ...**
>
> We will fix the typos and double-check our spelling and grammar in our manuscript, and accept your suggestion of adding the corresponding discussions in our main paper.
>
> **W3. The significance is strictly limited by the necessity of a good pre-trained ...**
>
> Our method is specifically designed for VLM prompt-tuning instead of classical neural networks. For large VLMs, pre-trained models with strong prior knowledge are usually available, and it is much more effective and economical to fine-tune a pre-trained model than
> to train one from scratch. Therefore, prompt-tuning for VLMs has been widely researched in recent years. Based on these practical demands, our method is a simple yet effective method with wide application fields.
>
> **W4. The algorithmic novelty is marginal. The method essentially ...**
>
>  The main contribution of our DSPT is that it can effectively suppress the learning process from mislabeled samples, while keeping the gradient from the useful samples **self-adaptively**. This mechanism is achieved by nullifying the gradients of overconfident samples and preserving gradients from ambiguous samples, which is fundamentally different  from LogitClip and Logit Norm, which performs underlearning on all samples with additional hyper-parameter.
> Therefore, our method should not be interpreted as a similar method to LogitClip. Instead, it is a selective noise suppression strategy with simple and uniform expression and self-adaptive capability designed for VLM prompt-tuning under various noise levels.
>
>
> **Q2. The current evaluation relies solely on CLIP with a ViT-B/16 encoder. Have you ...**
>
>  We use the CLIP with ViT-B/16 because it is one of the classical settings for VLM prompt-tuning problems, which is also introduced in NLPrompt. To address the reviewer's concerns, we conduct an extra experiment by changing our ViT-B/16 to a larger VIT-L/14-336 backbone and test our method on the DTD dataset compared with the standard CoOp prompt tuning and NLPrompt. The performance of the new model, as presented in the following table, is consistent with that of the VLM with ViT-B/16 in our paper.
>
> | DTD-sym | 40%   | 60%   | 80%   | DTD-flip | 20%   | 30%   | 40%    |
> |---------|-------|-------|-------|----------|-------|-------|--------|
> | CoOp    | 65.74 | 58.17 | 34.36 | CoOp     | 67.10 | 58.09 | 48.52  |
> | NLPrompt      | 75.30 | 67.40 | 57.35 | NLPrompt       | 76.86 | 73.52 | 72.24  |
> | DSPT    | 75.77 | 70.89 | 60.96 | DSPT     | 77.64 | 73.81 | 75.53  |
>
> **Q3. How does the method distinguish between a genuinely mislabeled sample and ...**
>
> Pre-trained VLMs, such as CLIP in our paper, make confident predictions largely because the sample is already in their prior knowledge. Therefore, for out-of-distribution data, the VLM is likely to produce ambiguous predictions. These samples will be learned in our approach, and the mislabeled ones in them will
> be gradient-suppressed after the model provides more confident predictions on them in the subsequent training epochs.
>
> **Q4. Table 5 demonstrate that DSPT underperforms on CIFAR-10 ...**
>
>  Our method is specifically designed for VLM prompt-tuning and is not suitable for single-modal tasks with models trained from scratch. To combine our method with the classical noisy label learning task,
> additional efforts are needed to adapt our approach with pre-trained mode, which is a complete different setting.

---

> > ### Author Rebuttal · Reviewer_Yh1U · 2026-04-01
> >
> > I thank the authors for their rebuttal and the additional experimental work.
> >
> > I can acknowledge that the discussion has helped to clarify matters and I can consider the majority of the originally raised concerns to be well addressed.

---

> > > ### Author Response · Authors · 2026-04-05
> > >
> > > We sincerely thank the reviewer for the highly constructive feedback and the confirmation that the concerns are adequately addressed. Your constructive suggestions on our model’s novelty compared to LogitClip and the evaluation across different backbones have been greatly helpful in strengthening our work. We are heartened by the reviewer's confirmation that all concerns are now fully resolved.

---

### Official Review · Reviewer_u1R7 · 2026-03-12

**Soundness:** 3
**Presentation:** 3
**Significance:** 3
**Originality:** 3
**Overall Recommendation:** 4
**Confidence:** 3

**Summary:**

This paper proposes Double-Softmax Prompt Tuning (DSPT), a hyperparameter-free loss modification for CLIP-style prompt tuning under label noise. The core idea is to apply a second softmax before cross-entropy so that high-confidence prediction–label mismatches receive intrinsically suppressed gradients, while more ambiguous samples still contribute tempered updates. The paper provides gradient-level analysis, bounded-loss and noisy-risk results, and extensive experiments on nine datasets under synthetic symmetric and pair-flip noise, showing strong robustness relative to several prompt-tuning and noisy-label baselines.

**Compliance With Llm Reviewing Policy:**

Affirmed.

**Final Justification:**

I thank the authors for their rebuttal and maintain my scores.

**Key Questions For Authors:**

1. Can the authors explain more clearly what it means that, under DSPT, the losses of correctly labeled and mislabeled samples stay at similar levels throughout training? In particular, does the elevated/flattened loss for correctly labeled samples indicate a desirable conservative adaptation effect, or could it also imply under-learning of clean samples?

2. Can the authors clarify the scope of Theorem 3.5 relative to the pair-flip experiments? Under the pair-flip definition in the paper, it seems that the theorem only covers \eta\le0.5, whereas Table 2 / Sec. 4.3 highlights the 80% pair-flip setting. Can the theory be extended to this regime, or should the empirical claim be stated more carefully?

3. Can the authors report results in the clean or low-noise regime (e.g., 0% / 10% noise)? This would help clarify whether DSPT is still helpful when labels are mostly reliable, and would make the practical trade-off of the method much clearer.

These questions are important to my assessment. If the authors can provide convincing clarification on Q1–Q3, I would likely raise my overall recommendation.

**Limitations:**

yes

**Strengths And Weaknesses:**

Strengths:
1. Simple and practically appealing method. The proposed DSPT is a very simple, drop-in loss modification for CLIP-style prompt tuning under label noise. It does not require extra modules, sample selection pipelines, or additional hyperparameter tuning, which makes the method easy to implement and potentially easy for the community to adopt.
2. Clear motivation and good alignment with pretrained VLM behavior. The paper is built around a reasonable and intuitive observation: under noisy labels, high-confidence prediction-label mismatches can produce disproportionately harmful gradients for pretrained CLIP models, and DSPT is explicitly designed to suppress these disruptive updates while keeping more informative updates.
3. Strong and fairly comprehensive empirical study. The experimental evaluation is a clear strength. The paper reports results on nine datasets under multiple synthetic noise settings, compares against several prompt-tuning and noisy-label baselines, and also includes further analyses on severe noise regimes, gradient behavior, and related robust-loss variants. Overall, the empirical evidence is fairly extensive for a paper with such a simple method.

Weaknesses:
1. The optimization implications of Figure 4 are not sufficiently explained. The paper interprets Figure 4 as evidence of robustness, showing that under DSPT the losses of correctly labeled and mislabeled samples remain close to each other, whereas CoOp exhibits a much larger separation. However, the paper does not really discuss what it means that the loss of correctly labeled samples is also kept relatively high/flat under DSPT, or what optimization trade-off this introduces. In particular, if DSPT flattens the loss landscape for both clean and noisy samples, then the same mechanism that suppresses noisy gradients may also make learning from correct samples more conservative. I think this missing discussion is an important weakness.
2. There is a mismatch between the asymmetric-noise theory and the paper’s strongest asymmetric-noise empirical claim. The theoretical guarantee under asymmetric noise appears to require conditions that, under the paper’s pair-flip definition, only cover pair-flip noise up to \eta\le0.5. However, one of the paper’s strongest asymmetric-noise experiments is the 80% pair-flip setting in Table 2 / Sec. 4.3. This does not invalidate the empirical result, but it weakens the alignment between the theory and the strongest experimental claim.
3. The experiments focus on moderate-to-heavy synthetic noise settings, but do not report comparisons in the fully clean or mildly noisy regime. Given that the paper explicitly emphasizes conservative adaptation and Figure 4 suggests a broad flattening effect on sample losses, it would be important to know whether DSPT remains beneficial, or incurs a trade-off, when labels are mostly correct.

---

> ### Author Rebuttal · Authors · 2026-03-30
>
> Thank you for your valuable and insightful feedback. In response, we will provide corresponding answers with additional experiments to address your concerns in detail.
>
> **W1 & Q1. The optimization implications of Figure 4 are not sufficiently explained. The paper interprets Figure 4 as evidence ...**
>
> I appreciate your reply. This comment is constructive that requires our detailed explanations. The theory behind our Figure 4 is that the lower bound of double-softmax cross-entropy loss is not 0, but instead $\log(1 + \frac{C - 1}{e})$ as discussed in our Proposition 1, where C is the number of classification classes. For the Caltech101 dataset with 100 classes, this bound is approximately 3.63, which matches the curves in Figure 4. (b) and (d).
>
> In addition, the incorrect-loss in Figure 4 (b) remains at the level of about 4.60 throughout the first 20 epochs. This is because the model is producing very few prediction changes on mislabeled samples, implying that it is not fitting to noisy patterns. On the other hand, the clean losses drop from approximately 3.80 in epoch 1 to nearly 3.68 in epoch 20, meaning that the model is making more accurate predictions on correctly labeled samples.
>
> The analysis above suggests that Figure 4 is strong evidence of our model's robustness and adaptiveness compared to standard cross-entropy loss under label noise.
> We will add this discussion about Figure 4 in our manuscript.
>
> **W2 & Q2.  There is a mismatch between the asymmetric-noise theory and the paper’s strongest asymmetric-noise empirical claim. ...**
>
>  As discussed in our appendix, we have that:
>
> $$ \mathcal{R}^T_{\mathcal{L}}(f^\star) - \mathcal{R}^T_{\mathcal{L}}(\tilde{f}^\star) \leq C\log(\frac{e+C-1}{1 + e^{-1}(C - 1)}) \cdot \mathbb{E}_{(\bf{x},y)} \sim \mathcal{P} _ {clean}[T _ {yy}] + \mathbb{E} _ {(\bf{x},y)} \sim \mathcal{P} _ {clean}[\sum _ {i \neq y} (T _ {yy}- T _ {yi})( \mathcal{L}(\tilde{f}^\star(\bf{x},i)) - \mathcal{L}(\bf{f}^\star(\bf{x},i)))]$$
>
> Note that this deduction can be acquired without the assumption of $\eta \leq 0.5$, where $\mathcal{L}$ is our double-softmax cross-entropy loss, and $f^\star$ and $\tilde{f}^\star$ are the global minimizer on clean and noisy data distribution, respectively.
> As every element in the confusion matrix $T$ and our loss $\mathcal{L}$ are all bounded, the upper bound of the discrepancy between the optimal classifiers under clean and noisy labels still exists, while assuming $\eta \leq 0.5$ provides a simpler expression.
>
> **W3 & Q3. The experiments focus on moderate-to-heavy synthetic noise settings, but do not report  ...**
>
> As discussed in [1], vision language models for prompt tuning are inherently resistant to label noise to some degree. Therefore, the main target of our experiment is to verify the effectiveness of our model under moderate and heavy noise settings in which the model can be overwhelmed by mislabeled samples.
>
> To address the reviewer's concern, we conduct additional experiments on OxfordPets and EUROSat datasets under entirely clean datasets, 20% symmetric noise, and 10% pair-flip noise following the noise ratio of our main experiment.
> The results in the following table prove that our method is still effective under minor noise compared to NLPrompt and CoOp. In addition, our method shows compatible performance compared to CoOp under cleaning datasets, which confirms that our training hyper-parameter is suitable and does not introduces over or under-training.
>
> |          | Dataset      | clean | 20%-sym | 10%-flip  |
> |----------|--------------|-------|---------|-----------|
> | CoOp     | OxfordPets   | 93.96 | 91.04   | 88.79     |
> | NLPrompt |              | 93.25 | 92.97   | 93.28     |
> | DSPT     |              | 93.74 | 93.16   | 93.65     |
> | CoOp     | EuroSAT      | 94.58 | 93.63   | 92.28     |
> | NLPrompt |              | 94.30 | 93.65   | 94.26     |
> | DSPT     |              | 94.50 | 94.43   | 94.34     |
>
> [1] Cheng-En Wu, Yu Tian, Haichao Yu, Heng Wang, Pedro Morgado, Yu Hen Hu, and Linjie Yang. Why Is Prompt Tuning for Vision-Language Models Robust to Noisy Labels? In Proceedings of the IEEE/CVF International Conference on Computer Vision, pages 15488–15497, 2023.

---

> > ### Author Rebuttal · Reviewer_u1R7 · 2026-04-03
> >
> > Thank the authors for their rebuttal and the additional experimental work.

---

> > > ### Author Response · Authors · 2026-04-05
> > >
> > > We sincerely thank the reviewer for the highly constructive feedback. We are encouraged by your positive suggestions approving our work’s simplicity and experimental extensiveness. In addition, your insightful comments (Q1-Q3) regarding loss curves, theoretical analysis, and additional experiments on minor losses were instrumental in strengthening the rigor of our work. We are heartened to reach a complete consensus on the technical quality of the paper, and we sincerely hope this positive assessment will be reflected in the final evaluation.

---

### Official Review · Reviewer_US54 · 2026-03-13

**Soundness:** 3
**Presentation:** 3
**Significance:** 3
**Originality:** 3
**Overall Recommendation:** 4
**Confidence:** 2

**Summary:**

This paper addresses the high sensitivity of CLIP prompt tuning to label noise and proposes a hyperparameter-free Double-Softmax Prompt Tuning (DSPT) method. The key idea is to suppress gradients from highly erroneous noisy samples while preserving informative updates. Experimental results show that this simple, plug-and-play design achieves state-of-the-art robustness on multiple noisy-label benchmarks, outperforming methods that rely on more complex architectures and manually tuned hyperparameters.

**Compliance With Llm Reviewing Policy:**

Affirmed.

**Final Justification:**

I thank the authors for their comprehensive and well-structured response. Overall, I find it addresses my primary concerns. I have raised the score accordingly.

**Key Questions For Authors:**

Please refer to the weakness part.

**Limitations:**

Yes

**Strengths And Weaknesses:**

Strength
+ The proposed method is simple, intuitive, and easy to understand.

+ The paper also provides a theoretical analysis.

+ The experiments are extensive and suggest that the method is effective.

Weakness

1. The proposed method also suppresses the gradients of correctly labeled samples. When clean samples are inherently difficult to learn, this may hurt performance. In this sense, DSPT appears somewhat similar to a smoothing trick, and it lacks the ability to adapt to different levels of data cleanliness.

2. The paper itself notes that the method is not optimal under moderate noise levels, which also suggests that DSPT is sensitive to the degree of noise and may not generalize well across datasets of varying quality.

3. In the experiments, DSPT is trained using the full training set, whereas some of the compared methods are designed for or evaluated under few-shot settings. Although the appendix explains the reason for this choice, it is still unclear whether this experimental setting is fully fair.

4. For the compared baseline methods that involve hyperparameters, it is unclear whether the chosen hyperparameter settings are optimal. This raises some concern about the fairness of the comparison.

---

> ### Author Rebuttal · Authors · 2026-03-30
>
> We are grateful for your constructive feedback. In response, we will provide corresponding answers with additional experiments.
>
> **W1. The proposed method also suppresses the . ...**
>
> Our method should not be interpreted as a simple trick similar to label smoothing, which performs regularization among all samples evenly. Instead, it is a selective noise suppression strategy with simple and uniform expression and self-adaptive capability designed for VLM prompt-tuning under various noise levels.
>
> As revealed in Theorem 3.2, the main contribution of our DSPT is that it can effectively suppress the learning process from mislabeled samples, while keeping the gradient from the useful samples. This is achieved by nullifying the gradient from samples with highly-confident predictions, but discordant with the noisy label, as these samples are likely to be mislabeled. On the other hand, gradients from ambiguous samples that receive low confidence predictions from the model are preserved automatically, magnifying the memorization of novel samples absent in the prior knowledge.
>
> It is also necessary to mention that golden noisy label learning method that can eliminate noisy influence entirely while keeping the clean set perfectly intact, is not practically accessible. Existing noisy label learning strategies, such as ours, are all built on the tradeoffs between clean and mislabeled samples.
>
> **W2. The paper itself notes that the method is not optimal ...**
>
> Thank you for your comment.  Our method can effectively outperform the SOTA method NLPrompt in other noise setting including extreme noise cases, which  is sufficient to demonstrates our method's superiority in most cases.
> In addition, our method is hyper-parameter free, which is an advantage compared to LogitNorm and label smoothing.
>
> **W3. In the experiments, DSPT is trained using the full ...**
>
> As discussed in our appendix, this work mainly focuses on noisy label learning rather than few-shot learning, and proceeding on the full training set instead of selecting several samples from each class introduces extra stability and is more suitable for our main task. In addition, performing few-shot and full training are essentially identical except the training set size.
>
> To address the reviewers' and readers' concerns about whether our method still works under few-shot training, we conduct additional experiments on DTD s dataset with the same noise settings as our main experiments. The results in the following table confirm that our method remains effective
> in few shot settings compared to the SOTA method NLPrompt, proving that our method is both robust and versatile.
>
> |          |   symmetric |       |       |  pair-flip |       |        |
> |----------|-----------------------|-------|-------|----------------------|-------|--------|
> |          |          40%                   | 60%   | 80%   | 20%                  | 30%   | 40%    |
> | CoOp     | 47.57                 | 38.09 | 20.41 | 55.79                | 48.42 | 40.39  |
> | NLPrompt | 60.06                 | 53.46 | 36.33 | 62.18                | 61.22 | 60.54  |
> | DSPT     | 60.14                 | 54.87 | 42.78 | 65.46                | 62.02 | 61.38  |
>
>
> **W4. For the compared baseline methods that involve hyperparameters ...**
>
>  The main comparative SOTA baseline in our experiment is NLPrompt, and we follow the same hyperparameter settings for NLPrompt as in its original paper. LogitNorm and label smoothing are designed for classical noisy label learning and over-confidence rather than our setting. We include these methods in our experiment not as comparative SOTA methods, but to maintain our paper's integrity, proving that our method is essentially different from them.
>
>
> In addition, LogitNorm and label smoothing are not as hyperparameter sensitive as LogitClip. For simple demonstrations, we conduct experiments on LogitNorm on DTD and OxfordPets datasets with different noises. As shown in the table, LogitNorm shows small performance fluctuation near the default setting $\tau=1$, but is still not matchable to NLPrompt and DSPT.
>
> | $\tau$             | 0.01  | 0.1   | 1     | 10    | 100    |
> |--------------------|-------|-------|-------|-------|--------|
> | DTD-40%sym         | 38.66 | 55.82 | 66.86 | 68.43 | 64.83  |
> | DTD-30%flip        | 40.06 | 55.14 | 66.5  | 65.2  | 57.05  |
> | OxfordPets-40%sym  | 86.07 | 88.68 | 90.75 | 90.17 | 88.32  |
> | OxfordPets-30%flip | 85.83 | 86.12 | 83.46 | 80.32 | 65.74  |
>
> We will publish our source code in the future for further explorations.

---

> > ### Author Rebuttal · Reviewer_US54 · 2026-04-01
> >
> > Thanks for the rebuttal. My concerns have been adequately addressed.

---

> > > ### Author Response · Authors · 2026-04-05
> > >
> > > We sincerely thank the reviewer for the highly constructive feedback and the confirmation that the concerns are adequately addressed. We are heartened by your positive feedback, as your constructive comments on our paper’s novelty and experimental rigor are invaluable for revising our final manuscript.

---

### Official Review · Reviewer_exD5 · 2026-03-16

**Soundness:** 3
**Presentation:** 3
**Significance:** 2
**Originality:** 3
**Overall Recommendation:** 4
**Confidence:** 3

**Summary:**

This paper studies label-noise-robust prompt tuning for vision-language models and proposes Double-Softmax Prompt Tuning (DSPT), a very simple loss-level modification that aims to suppress harmful gradients induced by mislabeled samples. This work improves the robustness of CLIP-style prompt tuning under noisy supervision while preserving the strong prior of the pretrained VLM.

**Compliance With Llm Reviewing Policy:**

Affirmed.

**Final Justification:**

I thank the authors for their comprehensive and well-structured response. Overall, I find that your rebuttal addresses my primary concerns.

**Key Questions For Authors:**

See in Weaknesses.

**Limitations:**

Yes.

**Strengths And Weaknesses:**

## **Strengths**
This paper proposes a hyperparameter-free double-softmax objective, along with theoretical discussion and strong empirical results across a range of noisy-label image classification benchmarks. The experimental section is extensive and generally well executed. The method is also appealingly simple.

## **Weaknesses**
1. While the empirical results are strong, the evaluation scope is still somewhat narrow relative to the paper’s claims. I'm wondering whether the method remains effective in more challenging transfer settings, such as cross-domain adaptation, distribution shift, fine-grained domain transfer, or more realistic noisy annotation pipelines.
2. In realistic downstream adaptation, noise is often instance-dependent, semantically structured, or correlated with domain shift. Without such experiments, it is unclear how much of the gain would transfer to practical noisy-label scenarios rather than benchmark-simulated corruption.
3. Most experiments use a single CLIP backbone (ViT-B/16) and a CoOp-like shared prompt setting.
4. It would strengthen the paper to test whether the same effect holds for other VLM backbones.

---

> ### Author Rebuttal · Authors · 2026-03-30
>
> We greatly appreciate your valuable feedback and have just posted our response to your comments. We hope our response adequately addresses your points and restores your confidence in our work.
>
> **W1. While the empirical results are strong, the evaluation scope is still ...**
>
> Thank you for your reply. Our DSPT method is specifically designed for prompt-tuning with noisy labels for vision language models. As introducing noisy labels is inevitable in real-world practices, and prompt-tuning is a popular and economical fine-tuning method for VLM, our method has a wide range of applications, whose effectiveness has been proven by our experiment.
>
> Relying on the model’s prior knowledge, our double-softmax mechanism can suppress the gradient from confident samples while encouraging the VLM to learn from ambiguous samples. This mechanism leverages the intrinsic inductive bias of pre-trained VLMs to distinguish between reliable knowledge and noisy labels without manual intervention, and is deeply connected to our problem setting of tuning a pre-trained VLM backbone model. Therefore, this strategy can be further adjusted to solve problems with similar settings beyond our original application field.
> In the next answer, we provide an experiment that includes a more realistic noise annotation pipeline.
>
> **W2. In realistic downstream adaptation, noise is often instance-dependent, semantically structured, or ...**
>
>  Instance-independent noise, such as symmetric and pair-flip noise, is the most commonly adopted noise pattern for label noise-related problems in the community, as it is simple and most measurable. Therefore, we follow this convention for reliable results and fair comparisons, which is adequate to prove our method's effectiveness. To address the concern of the reviewer, we conduct experiments on DTD and OxfordPets datasets with 40%, 60%, and 80% rates of instance-dependent noise [1]. The result in the following table shows that our DSPT method still remains effective on more complex noises compared to NLPrompt and CoOp.
>
> | DTD      | 40%   | 60%   | 80%  | OxfordPets | 40%   | 60%   | 80%    |
> |----------|-------|-------|------|------------|-------|-------|--------|
> | CoOp     | 60.91 | 52.19 | 26.7 | CoOp       | 87.12 | 75.49 | 42.46  |
> | NLPrompt | 66.7  | 62.59 | 46.6 | NLPrompt   | 90.95 | 90.88 | 83.58  |
> | DSPT     | 68.27 | 63.38 | 52.3 | DSPT       | 93.21 | 92.07 | 89.85  |
>
> **W3 & 4. Most experiments use a single CLIP backbone (ViT-B/16) and a CoOp-like shared prompt setting. It would ..**
>
>  We use the CLIP following with ViT-B/16 and CoOp tuning as our backbone model because it is one of the classical settings for VLM prompt-tuning problems. To address the reviewer's concerns of our method's adaptivity on different VLM backbones, we conduct an extra experiment by changing our ViT-B/16 to a larger VIT-L/14-336 backbone and test our method on the DTD dataset compared with the standard CoOp prompt tuning and NLPrompt. The performance of the new model, as presented in the following table, is consistent with that of the VLM with ViT-B/16 in our paper.
>
> | DTD-sym | 40%   | 60%   | 80%   | DTD-flip | 20%   | 30%   | 40%    |
> |---------|-------|-------|-------|----------|-------|-------|--------|
> | CoOp    | 65.74 | 58.17 | 34.36 | CoOp     | 67.1  | 58.09 | 48.52  |
> | NLPrompt      | 75.3  | 67.4  | 57.35 | NLPrompt       | 76.86 | 73.52 | 72.24  |
> | DSPT    | 75.77 | 70.89 | 60.96 | DSPT     | 77.64 | 73.81 | 75.53  |
>
> [1]Xia, X., Liu, T., Han, B., Wang, N., Gong, M., Liu, H., Niu,G.,Tao, D., and Sugiyama, M. Part-dependent label noise:Towards instance-dependent label noise. Advances in Neural Information Processing Systems, 33:7597–7610, 2020.

---

> > ### Author Rebuttal · Reviewer_exD5 · 2026-04-01
> >
> > I thank the authors for their comprehensive and well-structured response. Overall, I find that your rebuttal addresses my primary concerns.

---

> > > ### Author Response · Authors · 2026-04-05
> > >
> > > We sincerely thank the reviewer for the highly constructive feedback and the time invested in reviewing our response. Your insightful comments on practical noise patterns and backbone architectures helped improve the quality of our work. We are heartened by the reviewer's confirmation that all concerns are now fully resolved.

---

### Decision · Program_Chairs · 2026-04-30

**Decision:**

Accept (regular)

**Comment:**

This paper focuses on improving noisy-label adaptation of pretrained VLMs. It proposes Double-Softmax Prompt Tuning (DSPT), a hyperparameter-free double-softmax loss for label-noise prompt tuning in CLIP, designed to suppress harmful gradients from mislabeled samples while preserving useful updates. Four experts in the field reviewed this paper, all rating it as weak accept. Reviewers agreed that the paper is clearly written and organized, generally well-motivated theoretically, and provides extensive empirical evaluations. The main weaknesses are the unclear analysis of the clean/low-noise levels and optimization behavior, and the limited evaluation scope beyond the standard CLIP and CoOp. Based on reviewer feedback and the authors’ satisfactory rebuttal, which addresses the most important concerns, with more experiments on instance-dependent noise, few-shot settings, and a larger backbone, I recommend it for acceptance. Authors should address issues raised by the reviewers in the final camera-ready version of the paper.